# CSL controls telomere maintenance and genome stability in human dermal fibroblasts

Giulia Bottoni [1,2,3,10], Atul Katarkar[3,10], Beatrice Tassone[3], Soumitra Ghosh[3], Andrea Clocchiatti[1,2], Sandro Goruppi[1,2], Pino Bordignon[3], Paris Jafari[1,3], Fabio Tordini[4,5], Thomas Lunardi[6], Wolfram Hoetzenecker[7], Victor Neel[8], Joachim Lingner[6] & G. Paolo Dotto[1,3,9]

Genomic instability is a hallmark of cancer. Whether it also occurs in Cancer Associated Fibroblasts (CAFs) remains to be carefully investigated. Loss of CSL/RBP-Jκ, the effector of canonical NOTCH signaling with intrinsic transcription repressive function, causes conversion of dermal fibroblasts into CAFs. Here, we find that CSL down-modulation triggers DNA damage, telomere loss and chromosome end fusions that also occur in skin Squamous Cell Carcinoma (SCC)-associated CAFs, in which CSL is decreased. Separately from its role in transcription, we show that CSL is part of a multiprotein telomere protective complex, binding directly and with high affinity to telomeric DNA as well as to UPF1 and Ku70/Ku80 proteins and being required for their telomere association. Taken together, the findings point to a central role of CSL in telomere homeostasis with important implications for genomic instability of cancer stromal cells and beyond.

[1] Cutaneous Biology Research Center, Massachusetts General Hospital, Charlestown, MA 02129, USA. [2] Department of Dermatology, Harvard Medical School, Boston, MA 02125, USA. [3] Department of Biochemistry, University of Lausanne, 1066 Epalinges, Switzerland. [4] Cancer Genomics Laboratory, Edo and Elvo Tempia Valenta Foundation, 13900 Biella, Italy. [5] Diatech Pharmacogenetics srl, 60035 Jesi, Italy. [6] Swiss Institute for Experimental Cancer Research, School of Life Sciences, Ecole Polytechnique Fédérale de Lausanne, 1015 Lausanne, Switzerland. [7] Department of Dermatology, Kepler University Hospital, 4020 Linz, Austria. [8] Department of Dermatology, Massachusetts General Hospital, Boston, MA 02114, USA. [9] International Cancer Prevention Institute, 1066 Epalinges, Switzerland. [10] These authors contributed equally: Giulia Bottoni, Atul Katarkar. Correspondence and requests for materials should be addressed to G.P.D. (email: paolo.dotto@unil.ch)

The skin is a model of major clinical significance for early steps of tumor development. Many cancer driver mutations are found in phenotypically normal epidermal tissues, pointing to the importance of concomitant stromal changes[1]. Conversion of dermal fibroblasts into Cancer Associated Fibroblasts (CAFs) can play a pivotal role in keratinocyte tumor development and field cancerization, a condition consisting of broad tissue alterations beyond sites of tumor formation that spread over time[2].

Genomic instability and resulting chromosomal aberrations are a key hallmark of cancer[3]. Whether or not they occur also in adjacent stroma remains a matter of debate. A number of studies have reported chromosome and/or gene copy number alterations in CAFs derived from breast, prostate, colorectal, and ovarian cancer[4–6]. However, these findings were questioned by others who raised the issue of technical artifacts[7,8]. Possible genetic changes in stromal fibroblasts need also to be considered in the context of inter- and intra-tumor heterogeneity, as well as different cancer types[9]. In this regard, studies on genomic integrity in dermal fibroblasts are important to conduct, given the persistent exposure of skin to exogenous clastogenic agents with high penetrating power, such as UVA, a main cause of tissue aging and cancer[10].

The ends of linear chromosomes are organized into telomeric nucleoprotein complexes with an essential protective function[11]. Maintenance of these complexes is required to prevent telomere loss, chromosomal end-to-end fusions, and extensive genomic instability[11]. These events trigger a DNA damage response (DDR) as a failsafe protective mechanism resulting in replicative arrest, senescence or apoptosis[12]. Cells with defective telomere maintenance can ultimately escape from proliferative arrest through induction of telomerase (TERT) expression and activity or alternative telomere lengthening[11,13,14].

A complex of six telomere-specific proteins (TRF1, TRF2, TPP1, POT1, TIN2, and RAP1), called shelterin, binds, and safeguards chromosome ends[15]. The telomere proteome is composed of additional telomere-associated proteins with a variety of functions[14]. Among these are Ku70 and Ku80, mostly known for their role in the classical non-homologous end joining (c-NHEJ) repair pathway, which they initiate by forming a ring around damaged DNA sites[16]. These proteins also associate with telomeres in a nontetramerized conformation, inhibiting alternative non-homologous end joining (alt-NHEJ) and homologous recombination (HR), and preventing loss of telomere repeats and chromosomal alterations, which are rapidly induced as a consequence of Ku70 and Ku80 deletion[14,17]. Another protein with protective function found at telomeres is UPF1 (Up-Frameshift Suppressor 1 Homolog)[18,19], an ATPase and RNA/DNA helicase well studied for its other role in various mRNA decay pathways[20]. While both Ku70/Ku80 and UPF1 have been shown to play a key role in telomere protection, the mechanisms that mediate their recruitment to telomeres remain to be explored.

CSL (RBP-Jκ) is a highly conserved DNA binding protein and effector of canonical NOTCH signaling, with an intrinsic transcriptional repressive function[21]. As in other cellular systems, in dermal fibroblasts NOTCH1 and CSL expression are inversely related as part of a reciprocal negative regulatory loop, with CSL downmodulation occurring in CAFs[22], as well as in dermal fibroblasts upon UVA exposure[23]. Loss of CSL repressive function in dermal fibroblasts triggers early steps of CAF activation, with induction of a large set of CAF effector genes together with p53-mediated cellular senescence as a failsafe protective mechanism, while concomitant loss of CSL and p53 promotes stromal and cancer cell expansion[22,24]. Here we report that, separately from its transcriptional function, CSL plays a key role in maintenance of genomic integrity in both dermal fibroblasts and CAFs. Mechanistically, CSL binds and anchors Ku70/Ku80 and UPF1 to telomeric DNA, orchestrating crucial aspects of telomere biology with relevant implications for tumor development.

## Results

### DNA damage in dermal fibroblasts and CAFs with CSL loss.
Deletion of the Csl gene in the mesenchymal skin compartment of mice results in dermal atrophy and fibroblast cell senescence already at birth, preceding the formation of inflammatory infiltrates and subsequent keratinocyte tumors[24]. Dermal fibroblast senescence and skin aging are also major consequences of UV exposure and ensuing DDR[25], which induce downmodulation of CSL expression[23]. Here we tested whether Csl loss is by itself sufficient to elicit a DDR, using levels of histone H2ax phosphorylation (γ-H2ax) as an indication[26]. Immunofluorescence analysis of the skin of newborn mice with dermal Csl deletion showed a strong increase of γ-H2ax positive dermal fibroblasts relative to those of control mice (Fig. 1a). Paralleling the in vivo findings, substantially higher γ-H2ax levels were found in dermal fibroblasts derived from mice (MDFs) with Csl deletion versus controls (Fig. 1b).

Silencing of the CSL gene in multiple strains of primary human dermal fibroblasts (HDFs) resulted also in γ-H2AX induction, with a markedly increased number of cells with genomic DNA breakage, as assessed by comet assays (Fig. 1c–e). Confirming the specificity of the results, γ-H2AX induction was strongly reduced by lentivirally-mediated CSL overexpression in HDFs with CSL gene silencing or treated with UVA, which, as previously reported[23], caused endogenous CSL downmodulation (Fig. 1f, g).

A connection between elevated DNA damage and loss of CSL was also found in clinical samples. In fact, immunofluorescence analysis of surgically excised skin samples showed increased γ-H2AX levels in fibroblasts adjacent to premalignant (actinic keratosis; AK) and malignant (SCC) cancer lesions, in both of which CSL levels are decreased[22,27], relative to fibroblasts of flanking unaffected skin (Fig. 2a, b and Supplementary Fig. 1a, b). Persistently increased γ-H2AX levels and augmented DNA breakage were also observed in multiple skin SCC-derived CAF strains, in which CSL expression is low[22,28,29], relative to matched HDFs of the same patients (Fig. 2c, d). In a dose-response experiment, γ-H2AX levels were strongly reduced in CAFs upon infection with an inducible CSL expression vector, even with a minimal CSL increase comparable with endogenous levels in HDFs (Fig. 2e and Supplementary Fig. 1c). Similarly, genomic DNA breakage was suppressed in CAFs with CSL overexpression (Fig. 2f).

Thus loss or downmodulation of CSL in dermal fibroblasts lead to DNA damage, which persists in CAFs and can be counteracted by increased CSL expression.

### Telomere loss and genome instability by CSL knockdown and in CAFs.
Telomeres are sites of great biological importance that are exposed to continuous DNA erosion/repair linked with replication[11]. Double immunofluorescence and fluorescence in situ hybridization (FISH) analysis of interphase nuclei and mitotic chromosome spreads showed a pronounced and preferential increase of γ-H2AX signal at telomeric versus non-telomeric regions in multiple HDFs strains upon CSL gene silencing, with similar signs of telomere DNA damage in MDFs with Csl deletion (Fig. 3a–c and Supplementary Fig. 2a).

qPCR analysis of genomic DNA with telomeric repeat-specific primers[30] showed a decrease of telomeric repeats in HDFs with silenced CSL (Fig. 3d). To test whether telomere loss was

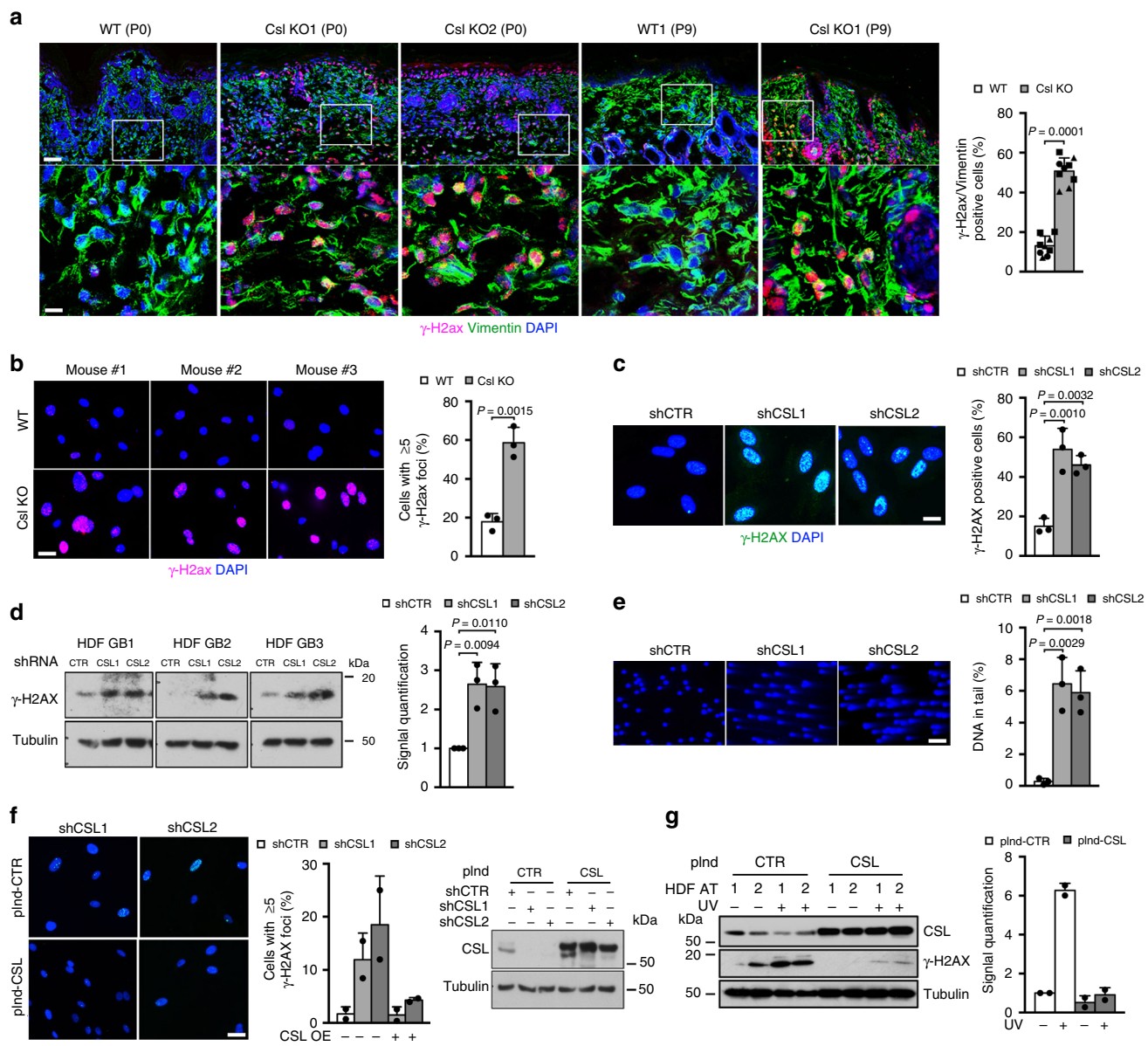

**Fig. 1** *CSL* loss induces DNA damage in mouse and human dermal fibroblasts. **a** γ-H2ax (magenta) and Vimentin (green) immunostaining of the skin of mice plus/minus mesenchymal *Csl* deletion (WT/KO) at the indicated days after birth (P0-9). Shown are representative low and high magnification images (scale bars, 100 and 10 μm) and quantification of double positive γ-H2ax and Vimentin cells. Circles, triangles, and squares represent P0, P6, and P9 mice, respectively. >100 Vimentin positive cells were counted in each case. $n$(WT P0) = 2, $n$(WT P6) = 3, $n$(WT P9) = 4, $n$(KO P0) = 2, $n$(WT P6) = 3, $n$(WT P9) = 4, ****$p$ = 0.0001, two-tailed unpaired $t$-test. **b** γ-H2ax immunostaining and quantification of early passage dermal fibroblasts derived from mice plus/minus mesenchymal *Csl* deletion (WT/KO). Scale bar, 10 μm. >300 cells were counted per sample. $n$(WT) = 3, $n$(KO) = 3, **$p$ < 0.01, two-tailed unpaired $t$-test. **c** γ-H2AX immunostaining and quantification of HDFs plus/minus infection with two *CSL* silencing lentiviruses versus empty vector control for 5 days. Scale bar, 5 μm. >245 cells were counted per sample. $n$(strain) = 3, **$p$ < 0.01, one-way ANOVA. **d** Immunoblot and densitometric quantification (after γ-TUBULIN normalization) of γ-H2AX protein levels in HDFs plus/minus *CSL* silencing as in **c**. $n$(strain) = 3, *$p$ < 0.05, one-way ANOVA. **e** Comet assays of HDFs plus/minus shRNA-mediated *CSL* silencing as in **c**. Scale bar, 20 μm. >40 cells were analyzed per sample. $n$(strain) = 3, **$p$ < 0.01, one-way ANOVA. **f** γ-H2AX immunostaining and immunoblot of CSL levels (with γ-TUBULIN normalization) in HDFs with *CSL* silencing and concomitant overexpression. HDFs stably infected with *CSL*-inducible lentiviral vector (pInd-CSL) or empty-vector control (pInd-CTR) were infected with two *CSL* silencing lentiviruses versus control for 5 days and concomitantly treated with doxycycline (500 ng ml$^{-1}$). Scale bar, 10 μm. >100 cells were counted per sample. $n$(strain) = 2. **g** Immunoblot and densitometric quantification (after γ-TUBULIN normalization) of γ-H2AX and CSL protein levels in two HDF strains (HDF AT1 and AT2) infected with an empty-vector control versus *CSL*-inducible virus plus/minus UVA treatment. After 5 days of doxycycline (500 ng ml$^{-1}$) treatment for CSL induction, HDFs were irradiated with UVA (0, 2 J cm$^{-2}$) and protein lysates were collected 6 h after exposure. $n$(strain) = 2. Bars represent mean ± SD

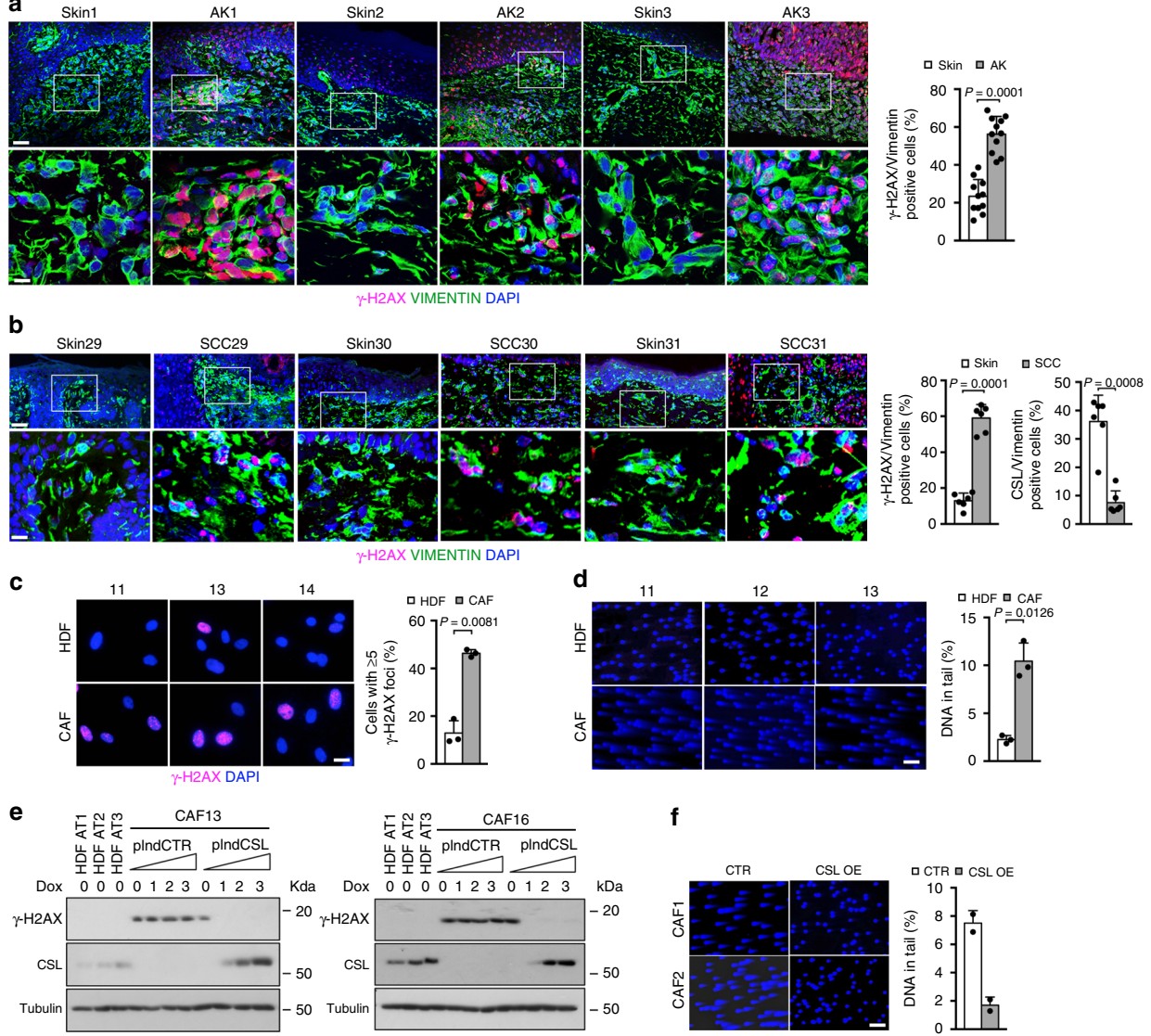

**Fig. 2** DNA damage induction in CAFs can be counteracted by CSL overexpression. **a** γ-H2AX (magenta) and VIMENTIN (green) immunostaining of AK underlying stroma versus flanking unaffected skin from multiple patients. Shown are representative low and high magnification images (scale bars, 50 and 10 μm) and quantification of double positive γ-H2AX and VIMENTIN cells. Decreased CSL expression and limited leukocytes infiltration were previously shown for five lesions[2] and for the remaining they were assessed by double immunostaining with anti-VIMENTIN and anti-CSL/anti-CD45 antibodies (Supplementary Fig. 1a). >120 VIMENTIN positive cells were counted per sample. $n$(AK/Skin) = 11, ****$p$ = 0.0001, two-tailed paired $t$-test. **b** γ-H2AX (magenta) and VIMENTIN (green) immunostaining of SCC underlying stroma versus flanking unaffected skin from multiple patients. Shown are representative low and high magnification images (scale bars, 50 and 10 μm) and quantification of double positive γ-H2AX and VIMENTIN cells. Decreased CSL expression and limited leukocytes infiltration were assessed by double immunostaining with anti-VIMENTIN and anti-CSL/anti-CD45 antibodies (right panel and Supplementary Fig. 1b). >77 VIMENTIN positive cells were counted per sample. $n$(SCC) = 6, $n$(Skin) = 6, ***$p$ < 0.001, two-tailed paired $t$-test. **c** γ-H2AX immunostaining of CAFs derived from three skin SCCs and matched HDFs from unaffected skin of the same patients. Scale bar, 10 μm. >175 cells were counted per sample. $n$(CAF strain) = 3, $n$(matched HDF strain) = 3, **$p$ < 0.01, two-tailed paired $t$-test. **d** Comet assays of three CAF and matched HDF strains. Scale bar, 50 μm. >135 cells were analyzed per sample. $n$(CAF strain) = 3, $n$(matched HDF strain) = 3, *$p$ < 0.05, two-tailed paired $t$-test. **e** Immunoblot of γ-H2AX and CSL protein levels (with γ-TUBULIN normalization) in two CAF strains infected with an inducible *CSL* overexpressing lentivirus versus empty vector control and treated with increasing concentrations of doxycycline for 5 days. 0, 1, 2, and 3 represent 0, 50, 200, and 500 ng ml$^{-1}$ of doxycycline, respectively. Three untreated HDF strains were analyzed in parallel as a reference. $n$(CAF strain) = 2. **f** Comet assays of two CAF strains infected with a constitutive *CSL* overexpressing retrovirus versus empty vector control for 5 days. Scale bar, 50 μm. > 303 cells were analyzed per sample. $n$(CAF strain) = 2. Bars represent mean ± SD

restricted to a subset of chromosomes or if it represented an overall shortening of telomeres, we employed telomere FISH analysis on chromosome spreads coupled with software-assisted image quantification, which showed only a slight reduction of FISH signal intensity in *CSL*-silenced HDFs (Fig. 3e and Supplementary Fig. 2b). This contrasted with a significant

increase of individual chromosomes with loss of one (OTL, one telomere loss) or two (TD, terminal deletion) telomeres in cells with silenced *CSL* (Fig. 3f and Supplementary Fig. 2c), suggesting that only a subset of chromosomes is affected. Two common consequences of telomere depletion are chromosome end joining and sister chromatid fusions[11], both of which were also markedly

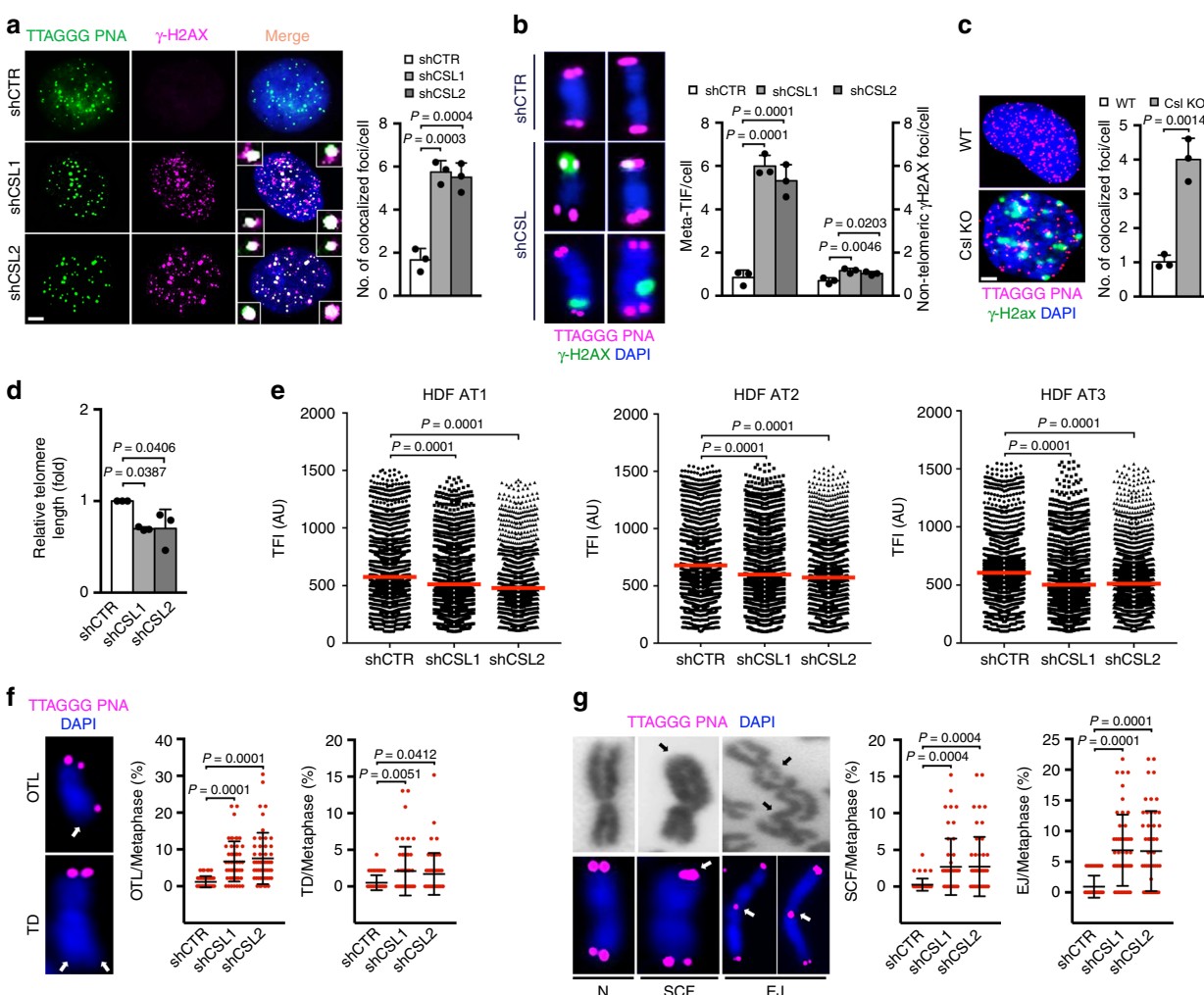

**Fig. 3** CSL depletion triggers telomere loss and genomic instability in HDFs and MDFs. **a** Telomeric DNA Q-FISH (TTAGGG PNA, green) and γ-H2AX immunostaining (magenta) colocalization signals (foci) in HDFs plus/minus *CSL* silencing (5 days). Scale bar, 1 μm. Fifty cells per sample were scored. $n$(strain) = 3, ***$p$ < 0.001, one-way ANOVA. **b** Telomeric DNA Q-FISH (magenta) and γ-H2AX immunostaining (green) colocalization signals (telomere dysfunction-induced foci, meta-TIF) and nontelomeric γ-H2AX foci in metaphase chromosome spreads from HDFs plus/minus *CSL* silencing. Additional images are in Supplementary Fig. 2a. Forty spreads per sample were scored. $n$(strain) = 3, *$p$ < 0.05, one-way ANOVA. **c** Telomeric DNA-FISH (magenta) and γ-H2ax immunostaining (green) colocalization signals (foci) in dermal fibroblasts from multiple mice plus/minus mesenchymal *Csl* deletion (WT/KO). Scale bar, 1 μm. >25 cells per sample were scored. $n$(WT) = 3, $n$(KO) = 3, **$p$ < 0.01, two-tailed unpaired $t$-test. **d** qPCR analysis of average telomere length normalized to the *ALBUMIN* gene in HDFs plus/minus *CSL* silencing. $n$(strain) = 3, *$p$ < 0.05, one-way ANOVA. **e** Analysis of telomere length by Q-FISH in HDFs plus/minus *CSL* silencing. Scatter plots show distribution of telomere fluorescence intensity (TFI) in arbitrary units (AU). >3000 telomeres were quantified per sample. Histograms with telomere length distribution are in Supplementary Fig. 2b. $n$(telomere) > 3000, $n$(strain) = 3, ****$p$ = 0.0001, one-way ANOVA. **f** Representative images of one telomere loss (OTL) and terminal deletion (TD) (white arrows) and quantification of the percentage of chromosomes carrying OTLs or TDs per metaphase in HDFs plus/minus *CSL* silencing. Additional images are in Supplementary Fig. 2c. Mean ± SD, $n$(spread) = 50, $n$(strain) = 2, *$p$ < 0.05, one-way ANOVA. **g** Representative phase contrast (top) and telomere FISH (bottom) images of normal chromosomes (N), chromosomes with sister chromatid fusion (SCF) or with chromosomal end joining (EJ) (arrows), and quantification of the percentage of chromosomes carrying SCFs or EJs per metaphase in HDFs plus/minus *CSL* silencing. Additional images are in Supplementary Fig. 2c. Mean ± SD, $n$(spread) = 50, $n$(strain) = 2, ***$p$ < 0.001, one-way ANOVA. Bars represent mean ± SD

increased in metaphase spreads from *CSL*-silenced HDFs versus controls (Fig. 3g and Supplementary Fig. 2c).

We previously showed that loss of CSL results in p53-dependent senescence as a failsafe mechanism against CAF activation[22]. As cellular senescence is also a trigger of telomere and genomic instability, we tested whether similar alterations occurred upon *CSL* silencing in *TP53* deficient cells that escape from proliferative arrest. Concomitant silencing of *CSL* and *TP53* in multiple HDF strains resulted in a similar induction of DNA damage at chromosome ends as in cells with *CSL* silencing alone, with a greater increase of chromosomes with telomere loss and sister chromatid fusions (Supplementary Fig. 3a–d).

Consistent with the findings in HDFs with silencing of *CSL* plus/minus *TP53*, DNA damage at chromosome ends, a slight decrease of telomeric repeats and a significant number of chromosomes with one or two telomeres loss, as well as chromosome fusions, were found in multiple CAF strains relative to matched HDFs from surrounding skin (Fig. 4a and Supplementary Fig. 4a, b).

An important question was the relationship between telomere instability and telomerase activity. No changes in *hTERT* were found in HDFs with *CSL* gene silencing alone, while both *hTERT* expression and activity were induced in HDFs with concomitant knockdown of *CSL* and *TP53* (Supplementary Fig. 4c, d).

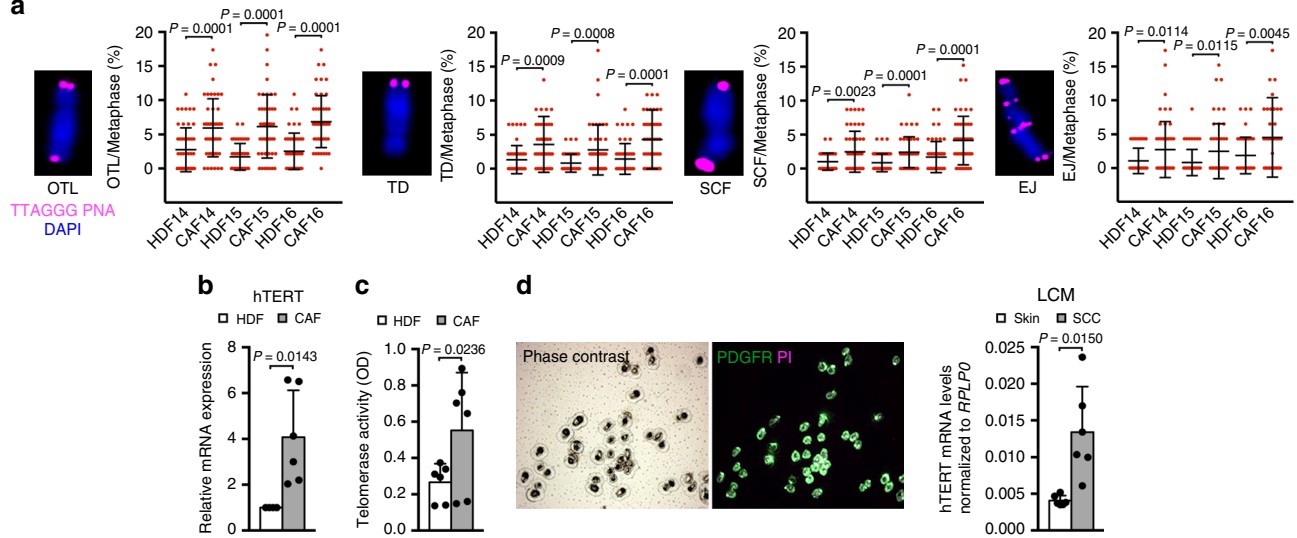

**Fig. 4** CAFs display persistent genomic instability and hTERT reactivation. **a** Representative images and quantification of the percentage of chromosomes carrying OTLs, TDs, SCFs, and EJs per metaphase in three CAF and matched HDF strains. Data are shown as in Fig. 3f, g. Mean ± SD, $n$(spread) = 47, $n$(CAF strain) = 3, $n$(matched HDF strain) = 3, *$p < 0.05$, two-tailed unpaired $t$-test. **b** RT-qPCR analysis of *hTERT* expression, normalized to *RPLP0*, in six CAF and matched HDF strains. $n$(CAF strain) = 6, $n$(matched HDF strain) = 6, *$p < 0.05$, two-tailed paired $t$-test. **c** Telomerase activity measured in the same six CAF and matched HDF strains as in **b**. $n$(CAF strain) = 6, $n$(matched HDF strain) = 6, *$p < 0.05$, two-tailed paired $t$-test. **d** RT-qPCR analysis of *hTERT* expression, normalized to *RPLP0*, of in situ SCCs, and matched unaffected skin sections (from which CAFs and HDFs in Fig. 4b were derived) processed for fluorescence-guided laser capture microdissection (LCM) utilizing anti-PDGFRα-FITC conjugated antibodies (green) and propidium iodide (PI, magenta) staining for nuclei identification. Shown are also representative images of cells captured on LCM caps. $n$(SCC CAF) = 6, $n$(matched HDF) = 6, *$p < 0.05$, two-tailed paired $t$-test. Bars represent mean ± SD

Importantly, increased *hTERT* expression and activity were also found in multiple CAF strains versus matched HDFs (Fig. 4b, c), with the finding being further validated in vivo, by fluorescence guided laser capture microdissection (LCM) of fibroblasts (PDGFRα-positive) associated with SCC lesions versus flanking skin (Fig. 4d).

Thus, decreased CSL expression in stromal fibroblasts leads to telomeric DNA damage, with loss and alterations of chromosome ends that persist in CAFs.

**CSL protects telomeres independently from gene transcription.** A number of indirect mechanisms related to CSL transcription regulatory function could underlie the above effects. As an alternative possibility, CSL could directly participate in DNA damage and telomere maintenance by binding to proteins involved in these processes, as suggested by initial mass spectrometry (MS) analysis of polypeptides coimmunoprecipitated with endogenous CSL in HDFs. In fact, co-immunoprecipitation (Co-IP) assays with several HDF strains showed that endogenous CSL associates with Ku70, Ku80, and UPF1, three proteins implicated in DNA repair, and telomere homeostasis[14,17,18,20] (Fig. 5a). By sequential immunoprecipitation with antibodies against CSL and UPF1 it was possible to recover Ku70/Ku80 proteins, pointing to their concomitant association (Fig. 5b). The findings were further confirmed by PLA assays, which showed the existence of nuclear complexes formed by CSL with both UPF1 and Ku70/Ku80 (Fig. 5c, d).

To assess whether CSL could associate directly with UPF1 and Ku70, we resorted to microscale thermophoresis (MST), a technique that enables determination of binding affinities of two molecules on the basis of their movement in aqueous solution as a function of temperature gradients[31]. We found that purified recombinant CSL, Ku70, and UPF1 proteins could directly

associate with each other, with CSL having a particularly elevated affinity for Ku70 (Fig. 5e and Supplementary Fig. 5a–d).

To address the role of the above complex, we first tested whether Ku70, Ku80, and UPF1 could participate in CSL transcription regulatory function, as a repressor of CAF effector genes[22]. However, while expression of CAF markers such as *IL6*, *ACTA2*, *POSTN*, or *FAP* was induced by *CSL* downmodulation, no such induction occurred upon *Ku70*, *Ku80*, or *UPF1* gene silencing (Fig. 5f and Supplementary Fig. 6a). RNA-seq analysis of HDFs with *CSL* versus *UPF1* gene silencing confirmed the finding for an extended number of CAF effector genes (Fig. 5g) and showed globally divergent effects on gene transcription (Supplementary Data 1). This contrasted with the similarity of effects caused by *CSL* and *UPF1* knockdown at telomere ends (Supplementary Fig. 6b–f).

Given the above results, an attractive possibility is that CSL could bind to telomeres together with its interactors. To further assess this hypothesis, we analyzed the ChIP-seq (chromatin immunoprecipitation combined with massively parallel DNA sequencing) profiles of endogenous CSL in HDFs[22]. High CSL binding peaks were found at telomeric regions overlapping with TRF1/TRF2 binding motifs on multiple chromosome ends (Fig. 6a and Supplementary Data 2). To validate these results, we employed a quantitative telomeric binding assay based on ChIP with antibodies against specific proteins followed by dot blot determination of telomeric DNA enrichment, using probes for Alu repeats as negative control of specificity. As shown in Fig. 6b and Supplementary Fig. 7a, CSL was found to bind to telomeres in multiple HDF strains as did the TRF1/TRF2 and UPF1 proteins. Our results were further extended by studies with HEK293T cells, showing that, even in this case, endogenous CSL binds to telomeres (Fig. 6c), while transfection of Flag-tagged CSL constructs highlighted that the β-trefoil region of CSL, which was reported to bind to the DNA minor groove[32], interacted strongly

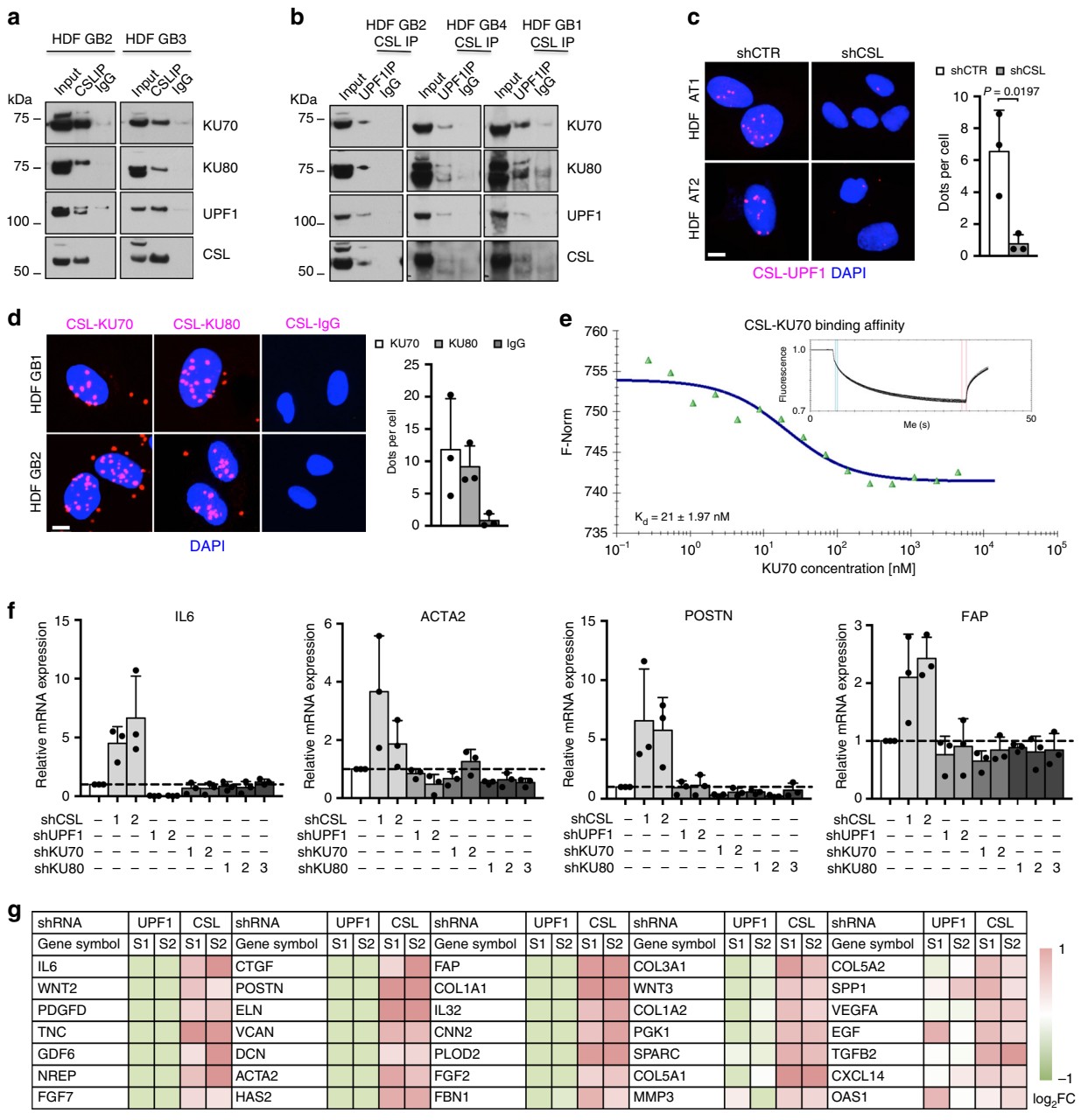

**Fig. 5** CSL binds to Ku70, Ku80, and UPF1 forming a multiprotein complex. **a** Co-immunoprecipitation (co-IP) analysis of HDFs with anti-CSL antibodies or non-immune IgG followed by immunoblotting with antibodies against the indicated proteins. $n(\text{strain}) = 2$. **b** Sequential co-IP analysis of HDFs with antibodies against CSL followed by IP with antibodies against UPF1 or non-immune IgG, and immunoblotting with antibodies against the indicated proteins. $n(\text{strain}) = 3$. **c** Proximity ligation assays (PLAs) of CSL and UPF1 association. HDFs with silenced *CSL* were used as negative control. Scale bar, 2 μm. Number of dots per cell was counted. $n(\text{cells}) > 43$ per condition, $n(\text{strain}) = 3$, *$p < 0.05$, two-tailed unpaired $t$-test. **d** PLAs of CSL and Ku70 and Ku80 association. Scale bar, 2 μm. Number of dots per cell was counted. $n(\text{cells}) > 54$ per condition, $n(\text{strain}) = 3$. **e** Binding of recombinant CSL and Ku70 proteins as measured by microscale thermophoresis (MST). Inset: thermophoretic movement of fluorescently-labeled CSL. Specificity controls are in Supplementary Fig. 5c, d. **f** RT-qPCR of CAF effector genes in HDFs plus/minus *UPF1/Ku70/Ku80* versus *CSL* gene silencing for 6 days. Silencing controls are in Supplementary Fig. 6a. **g** RNA-seq analysis of CAF effector genes in HDFs plus/minus *UPF1* versus *CSL* gene silencing for 7 days. Heatmap of differentially expressed genes in HDFs with *CSL* or *UPF1* silencing relative to control is in log2 scale. Bars represent mean ± SD

with telomeres as compared with its C-terminal and N-terminal domains (Fig. 6d and Supplementary Fig. 7b).

MST was also used to assess whether CSL could directly bind to telomeric DNA repeats devoid of canonical CSL recognition sites and in the absence of ancillary proteins. We found that purified recombinant CSL bound to telomeric DNA with only two-folds lower affinity than to its previously established DNA recognition sequence[33] ($K_d = 232–235$ nM versus 121 nM) and

with ten-folds higher affinity than to scrambled DNA ($K_d = 3380$ nM) (Fig. 6e, f and Supplementary Fig. 7c, d, f). While TRF2 was found to bind to telomeric DNA with very high affinity ($K_d = 12.2$ nM), UPF1 and monomeric Ku70 showed no binding specificity ($K_d = 3520$ and 2370 nM, respectively; Supplementary Fig. 7e, g, h).

Thus, CSL can bind to telomeric DNA repeats with high affinity.

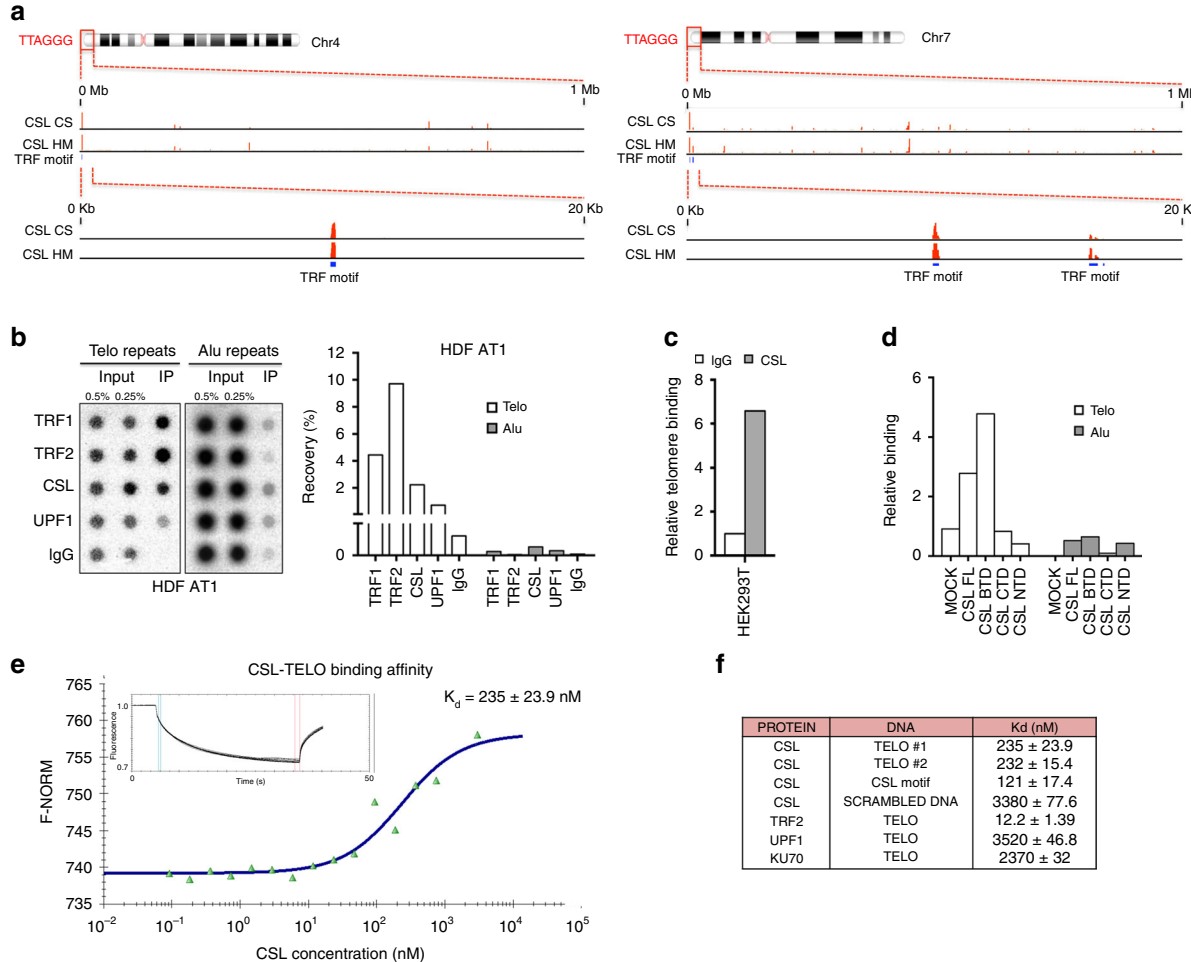

**Fig. 6** CSL binds to telomeres. **a** Position of CSL binding peaks (red) at telomeric ends of two chromosomes (chr4 and chr7) as revealed by Chromatin IP combined with DNA sequencing (ChIP-seq) analysis with two different anti-CSL antibodies (CS, Cell Signaling; HM, HomeMade)[22]. Chromosomal regions of CSL binding are shown at two different scales together with the predicted position of TRF binding motifs (blue bars). Complete list and position of CSL binding sites at chromosome ends detected by ChIP-seq analysis in HDFs is provided in Supplementary Data 2. **b** Densitometric quantification of CSL binding to telomeres by ChIP assays of a HDF strain (AT1) with antibodies against the indicated proteins in parallel with non-immune IgG followed by DNA dot blot hybridization with probes detecting telomeric (Telo) or Alu repeats. Similar experiments with two other HDF strains (AT2 and AT3) are in Supplementary Fig. 7a. **c** Telomere binding assay by ChIP/qPCR with antibodies against endogenous CSL, in parallel with non-immune IgG, in HEK293T cells. **d** Telomere binding assays of HEK293T cells expressing FLAG-tagged CSL full length (FL), CSL BTD, CSL CTD, or CSL NTD domains by ChIP/qPCR with anti-FLAG antibodies followed by qPCR with telomere- and alu-specific primers. Non-immune IgGs were used for normalization. A second independent experiment is shown in Supplementary Fig. 7b. **e** Binding of recombinant CSL protein to telomeric repeat DNA (TELO) as measured by MST. Inset: thermophoretic movement of fluorescently-labeled TELO. A second independent experiment and specificity controls are shown in Supplementary Fig. 7c–f. **f** Summary table of dissociation constants ($K_d$) of the indicated proteins from TELO DNA as determined by MST assays. SCRAMBLED DNA was used as specificity control

**CSL anchors the UPF1, Ku70, and Ku80 proteins to telomeres.** To further explore the functional significance of the findings, results were expanded by ChIP assays with antibodies against various proteins. Parallel immunoprecipitation of CSL and UPF1, followed by determination of recovered telomeric DNA by qPCR[34,35], confirmed specific binding of the two proteins to telomeric regions and sequential immunoprecipitation showed concomitant binding of the two (Fig. 7a).

To assess whether CSL may serve as anchor for the UPF1 and Ku70/Ku80 proteins, we resorted to *CSL* gene silencing for a restricted period of time (3 days), not sufficient to cause a reduction of telomere repeats (Supplementary Fig. 8a). As shown in Fig. 7b and Supplementary Fig. 8b, binding of UPF1, Ku70, and Ku80 to telomeres was drastically reduced by *CSL* knockdown, while binding of TRF1 and TRF2 was unaffected (Fig. 7c and Supplementary Fig. 8c).

The compromised binding of UPF1, Ku70, and Ku80 to telomeres in HDFs with *CSL* gene silencing was rescued by CSL overexpression (Fig. 7b and Supplementary Fig. 8b). Even in HDFs with basal CSL levels, overexpression of this protein resulted in increased recruitment of UPF1, Ku70, and Ku80 to telomeres (Fig. 7d and Supplementary Fig. 8d). The findings were further confirmed in HEK293T cells, in which increased CSL expression enhanced recruitment of Ku70 to telomeres in a dose dependent manner (Fig. 7e and Supplementary Fig. 8e).

For an in vivo validation of the results, we resorted to PLA assays with anti-shelterin antibodies (TRF1 and TRF2) as a proxy for detection of chromosome ends. Immunofluorescence analysis of skin SCC lesions versus surrounding unaffected skin showed that only expression of CSL (Fig. 2b) but not UPF1, Ku70, and Ku80 (Supplementary Fig. 8f) were decreased in the SCC-associated fibroblasts. However, while association of all these

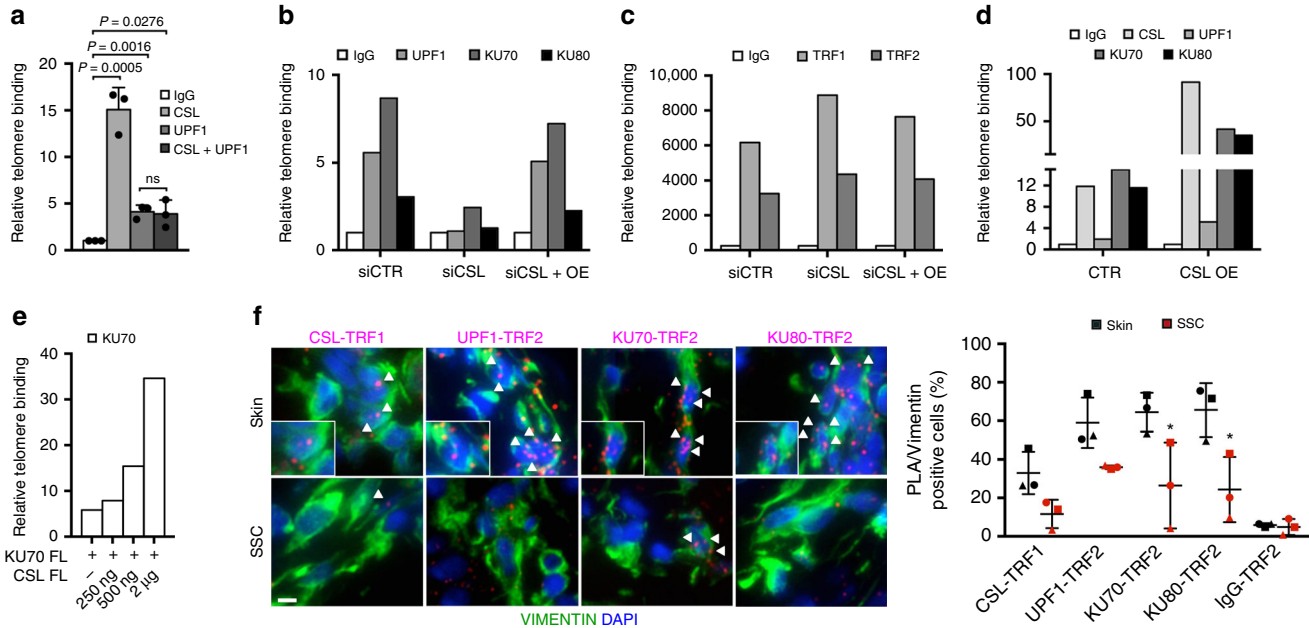

**Fig. 7** UPF1/Ku70/Ku80 recruitment to telomeres is impaired by *CSL* loss. **a** Telomere binding assay by ChIP/qPCR with antibodies against CSL and UPF1, individually or sequentially (CSL+UPF1), in parallel with non-immune IgG, in HDFs as in[34,35]. *n*(strain) = 3, *p < 0.05, two-tailed unpaired *t*-test. **b** Telomere binding assays by ChIP/qPCR with antibodies against the indicated proteins in HDFs (GB1) plus/minus siRNA-mediated *CSL* silencing (3 days) or *CSL* silencing and concomitant lentivirally induced *CSL* overexpression (OE). Similar experiments with two additional HDF strains (GB3 and GB4) are in Supplementary Fig. 8b. **c** Telomere binding assays by ChIP/qPCR analysis of the same cells as in **b** with antibodies against TRF1/TRF2. Similar experiments with two additional HDF strains (GB3 and GB4) are in Supplementary Fig. 8c. **d** Telomere binding assays by ChIP/qPCR with antibodies against the indicated proteins in HDFs (GB1) plus/minus lentivirally induced *CSL* overexpression (OE) for 3 days. Similar experiments with two additional HDF strains (GB3 and GB4) are in Supplementary Fig. 8d. **e** Telomere binding assays by ChIP/qPCR with anti FLAG-tag antibodies in HEK293T cells expressing increasing amounts (0, 250 ng, 500 ng, and 2 μg) of FLAG-tagged full length (FL) CSL together with full length (FL) Ku70 (2 μg). Non-immune IgGs were used for normalization. A second independent experiment is in Supplementary Fig. 8e. **f** PLAs of stromal fibroblasts (identified by VIMENTIN staining) from unaffected skin versus flanking SCC with TRF1 or TRF2 antibodies in combination with antibodies against the other indicated proteins. Scale bar, 5 μm. Quantification of CSL and UPF1/Ku70/Ku80/TRF1/TRF2 levels in the same samples are in Fig. 2b and Supplementary Fig. 8f, respectively. Triangles, circles, and squares point to values from flanking skin (black) and corresponding SCC (red) from three patients. Non-immune IgGs were used as control. Mean ± SD, *n*(cells) > 77 per condition, *n*(SCC) = 3, *n*(matched Skin) = 3, *p < 0.05, two-tailed paired *t*-test. Bars represent mean ± SD

proteins with TRF1 or TRF2 was readily detectable in dermal fibroblasts of unaffected skin, their binding in the SCC stromal fibroblasts was consistently decreased (Fig. 7f).

Thus, CSL is essential for effective anchoring of UPF1, Ku70, and Ku80 to telomeres.

**CSL provides a handle for Ku70 and Ku80 binding to telomeres.** For further molecular insights, we mapped the regions of mutual interactions by Co-IP of epitope-tagged proteins in HEK293T cells. Co-IP experiments were carried out in cells co-expressing CSL and epitope-tagged truncations of the Ku70 protein, lacking one or more of its N-terminus (N-Ku, aa 1–257), core (Core-Ku, aa 262–464), C-terminus (C-Ku aa 464–560), and SAP (aa 573–609) domains[36,37]. As expected from previous work[38], binding to Ku80 was detected only with a Ku70 fragment retaining the C-Ku and SAP domains (aa 257–609) (Fig. 8a). CSL association was found with the latter truncation and an additional one retaining most of the protein but lacking the SAP domain (aa 1–573) (Fig. 8a). CSL binding to the C-Ku domain was further confirmed by Co-IP assays with an additional derivative retaining this region and the SAP domain only (aa 464–609) (Fig. 8b).

CSL consists of three domains: NTD (N-terminal domain), BTD (β-trefoil domain), and CTD (C-terminal domain). Of these, the central β-trefoil domain of CSL (BTD, aa 166–334), with a key role in DNA recognition, as well as transcription complex formation[39], was found to associate with both Ku70 and UPF1

(aa 166–487) as effectively as the full-length protein (Fig. 8c and Supplementary Fig. 9a). We further evaluated the consequences of five single amino acid substitutions in the CSL BTD that were previously reported, at the equivalent position in the mouse protein, to suppress CSL binding to DNA (R192H)[40] or its transcription complex associated proteins (F235R, V237R, A258R, and Q307R)[41–46] (as summarized in Supplementary Fig. 9b). As shown in Fig. 8d, binding to UPF1 was unaffected by these mutations, while the association with Ku70 was specifically compromised by only two residues (R192H and A258R), pointing to the specificity of the interactions.

To integrate the above results, protein docking analysis was used to engender a 3D model of CSL, Ku70, and Ku80 interactions and their binding to telomeric DNA. We first docked telomeric DNA to the DNA binding pocket of CSL, choosing the top ranked configuration with minimal deviation from the crystallographically determined CSL-DNA complex[47]. We then docked Ku70 in its 3D configuration retrieved from the Swiss model server (https://swissmodel.expansy.org/) selecting the top ranked model of interactions with CSL on the basis of their experimentally determined binding regions. Irrespectively of whether we docked the whole Ku70 protein, the Ku70/Ku80 heterodimer or the Ku70 C-Ku/SAP domains only, the same two specific stretches of Ku70 were found to bind with highest rank to the CSL-DNA complex, connected by a flexible loop, with no direct interactions with Ku80 (Fig. 8e, f and Supplementary

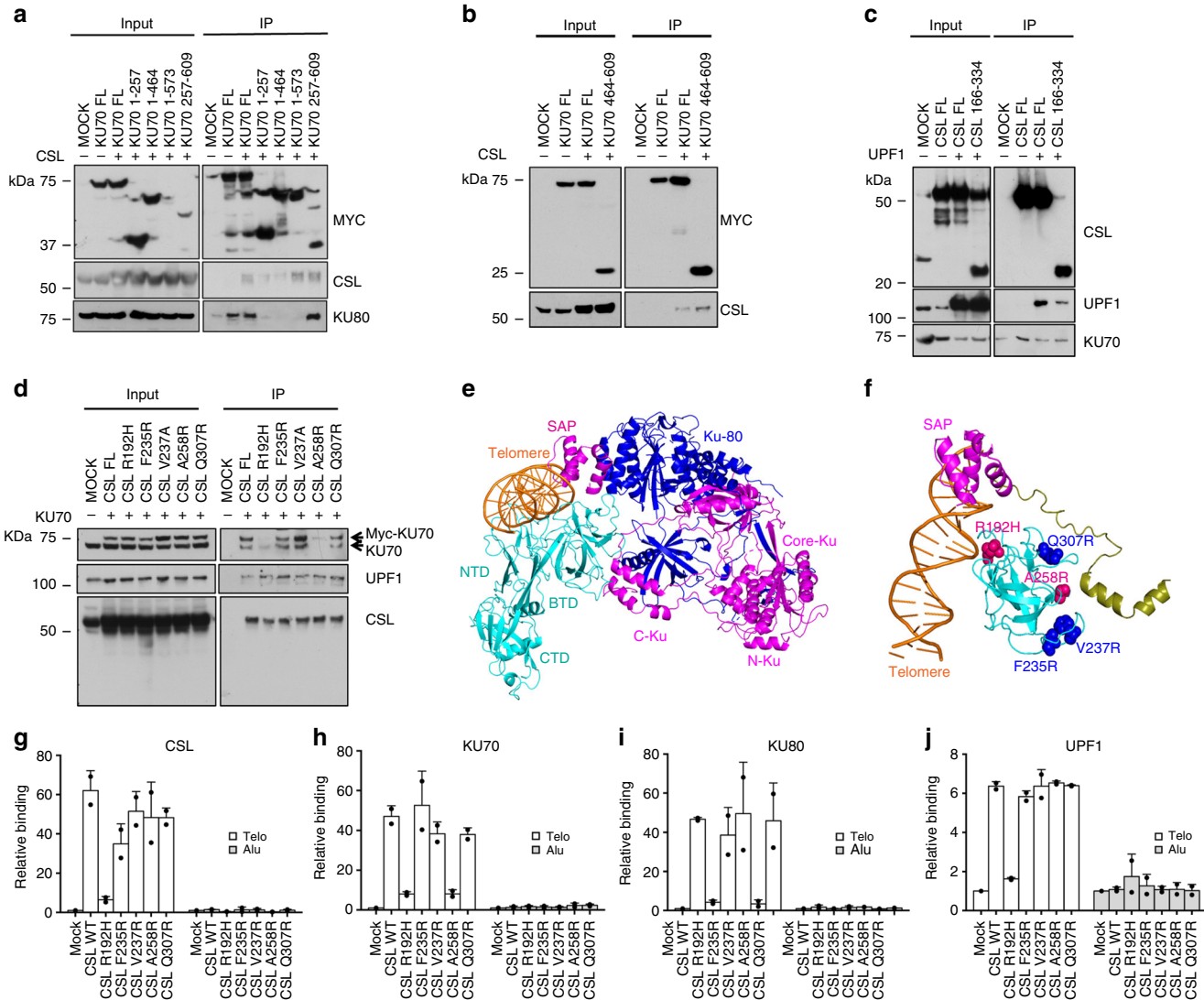

**Fig. 8** Mapping, mutagenesis and docking analysis of CSL/UPF1/Ku70/Ku80 interaction. **a** Co-IP analysis of HEK293T cells expressing MYC-tagged full length (FL) Ku70 and 1–257, 1–464, 1–573, or 257–609 domains plus/minus full length (FL) CSL with anti-MYC magnetic beads followed by immunoblotting with antibodies against the indicated proteins. **b** Co-IP analysis of HEK293T cells expressing MYC-tagged full length (FL) Ku70 or its 464–609 domain plus/minus full length (FL) CSL with anti-MYC magnetic beads followed by immunoblotting with antibodies against the indicated proteins. **c** Co-IP analysis of HEK293T cells expressing FLAG-tagged full length (FL) CSL and CSL BTD (166–334) domains plus/minus full length (FL) UPF1 with anti-FLAG magnetic beads followed by immunoblotting with antibodies against the indicated proteins. A second independent experiment is in Supplementary Fig. 9a. **d** Co-IP analysis of HEK293T cells expressing FLAG-tagged full length (FL) CSL and CSL point mutants (R192H, F235R, V237R, A258R, and Q307R) plus/minus full length (FL) Ku70 with anti-FLAG magnetic beads followed by immunoblotting with antibodies against the indicated proteins. Additional information on CSL point mutants is in Supplementary Fig. 9b. **e** Cartoon representation showing the docking complex between CSL (cyan)—Telomere DNA (orange)—Ku70 (magenta) and Ku80 (blue) using HDOCK server. Ku70 is shown to interact with CSL-BTD domain and bind to CSL bound telomere DNA through its SAP domain, while Ku80 binds indirectly through Ku70. **f** Close up view of the docking complex between CSL—Telomere DNA—Ku70 showing the α-helix of C-Ku domain (olive) interacting with CSL-BTD domain (cyan), and Ku70 SAP domain (magenta) interacting with telomere DNA (orange). The "hot spot" mutations that abrogate CSL-Ku70 interaction and Ku70-telomeric DNA association as in **d** are shown in pink (A258R and R192H). Additional non-interfering mutations are labeled in blue (F235R, V237R, and Q307R). **g–j** Telomeric binding assays with antibodies against the indicated proteins followed by qPCR with telomere- and alu-specific primers in HEK293T cells expressing CSL full length (WT) and point mutants (R192H, F235R, V237R, A258R, and Q307R). Non-immune IgGs were used for normalization. Bars represent mean ± SD

Movie 1). The stretch of amino acids corresponding to the α-helix of the Ku70 C-Ku domain (aa 520–529) is predicted to interact with the CSL BTD domain (aa 256–259), which includes the A258 residue whose A-R substitution abrogates Ku70 binding. The second Ku70 region, corresponding to the SAP domain (aa 573–609), is predicted to interact with the CSL-bound telomeric DNA rather than CSL itself.

The above model suggests that Ku70 is anchored to telomeric DNA by CSL through two parallel mechanisms, one involving direct docking between the two proteins, which is abrogated by the CSL A258R substitution, and the other one through the intermediation of telomeric DNA, whose binding to CSL is abrogated by the R192H substitution. This model was experimentally tested by assessing the binding of the complex to

telomeric DNA in HEK293T cells overexpressing wild-type CSL or multiple point mutants. ChIP with anti-CSL antibodies followed by determination of recovered telomeric DNA by qPCR, showed that CSL binding to telomeres was unaffected by all single amino acid substitutions except the one (R192H) predicted to abrogate DNA binding (Fig. 8g). Supporting the 3D modeling results, ChIP of the same cell extracts with antibodies against Ku70 showed that binding of this protein to telomeric DNA was drastically decreased in cells expressing the DNA binding defective CSL mutant (R192H), as well as the CSL mutant with the A258R amino acid substitution predicted to be involved in direct CSL–Ku70 interactions (Fig. 8h). Telomere binding of Ku80 was similarly affected by the two CSL mutations, as could be expected from the Ku70/Ku80 heterodimer association[14] (Fig. 8i). The binding of UPF1 to telomeres was also dependent on CSL and abrogated in cells expressing the DNA binding defective CSL mutant (R192H). However, it was not affected CSL mutants with the other single point substitutions including the one impairing Ku70/Ku80 binding (Fig. 8j).

Thus, CSL provides a specific docking mechanism for Ku70 binding to telomeres that is also required for Ku80 recruitment, while UPF1 is anchored by CSL to telomeres through a different set of interactions.

## Discussion

Multiple signaling pathways converge on epigenetic and gene expression changes resulting in CAF activation[48]. In parallel, the possible contribution of genetic instability in stromal fibroblasts to CAF conversion warrants further investigation. This is of relevance for the cancer-stromal cell co-evolution process and expansion of lesions over time. CSL is a highly studied DNA binding protein known for its role in transcription, which functions as a key negative regulator of CAF-effector genes. Here we have unveiled a role of CSL, independent from transcription, as an essential determinant of genomic stability in both dermal fibroblasts and CAFs. CSL prevents DNA damage and loss or fusion of chromosomal ends, localizing at telomeres and functioning as anchor for other proteins with essential telomere protective functions.

Down-modulation of CSL expression, as it can be triggered by pro-carcinogenic insults such as UVA exposure[23], is sufficient to trigger early steps of CAF activation, with up-regulation of a large battery of tumor promoting CAF-effector genes[22,27]. In parallel, we have found that CSL loss leads to DNA damage and a strong DDR in both cultured dermal fibroblasts and in the intact skin. The DDR triggered by CSL depletion was mostly localized at telomeres. Rather than an overall shortening of telomeres, this was accompanied by extensive alterations of individual chromosomes, with loss of one (OTL, one telomere loss) or two (TD, terminal deletion) telomeres, chromosomes end joining and sister chromatid fusions. HDFs with concomitant loss of CSL and TP53, displayed an even greater induction of telomeric instability despite the escape from cellular senescence. Furthermore, persistent DNA damage and chromosomal alterations were found in CAFs, which exhibited a gain of hTERT expression and activity that may be required for their sustained proliferation. Consistent with a model of multistep CAF activation that we previously proposed[22], a similar hTERT induction was found in dermal fibroblasts with combined CSL and TP53 loss, rather than in cells with loss of CSL alone. Despite the sustained accumulation of genetic alterations, CAFs do not display a transformed phenotype, unlike cancer cells. Multiple reasons might account for the lack of transformation in CAFs, such as the limited proliferative potential of fibroblasts as compared with other cell types that are constantly subjected to renewal, the absence of concomitant aberrations that need to co-occur in order to establish a selective advantage, the presence of suppressive mechanisms that limit clonal growth, and the lack of extensive aneuploidy that is a hallmark of cancer.

Several convergent mechanisms can contribute to the genomic instability associated with conversion of stromal fibroblasts into CAFs. Among these, TGFβ signaling, which functions as an important trigger of CAF activation[48,49], has been reported to modulate genomic instability in a variety of cell types, including normal fibroblasts, through unrepaired DNA strand breaks and differential expression of DDR genes[50,51]. CSL silencing elicits global changes in gene expression that overlap, in part, with those triggered by TGFβ[22,27], suggesting that modulation of DDR genes and other indirect mechanisms related to the CSL transcription regulatory function might also play a relevant role in preserving genomic stability. However, separate from transcription, we have shown that the CSL protein directly participates in maintenance of chromosomal ends, as part of a telomere protective complex comprising the Ku70/Ku80 and UPF1 proteins. Silencing of Ku70, Ku80, or UPF1 did not elicit the same effects as loss of CSL on gene expression, indicating that these proteins do not participate in the CSL transcriptional regulatory function. Consistent with a separate role of CSL in telomere stability, we found that purified CSL is capable of binding telomeric DNA repeats directly, with comparable affinity as to its canonical CSL binding motif, while monomeric Ku70 and UPF1 bound to telomeric DNA with very low affinity under similar conditions. Further evidence discussed below indicates that CSL is essential for effective binding of the latter proteins to telomeres.

Mostly known for their role in the c-NHEJ pathway, Ku70 and Ku80 form a constitutive heterodimer with the outward surface composed mostly of Ku70 residues and a central ring structure, with prevalent Ku80 residues, that can circle DNA with high affinity through sequence-independent interactions[52]. The Ku dimer is the first player in the c-NHEJ reaction, bridging two free double-stranded DNA ends through the formation of a tetramer, which is followed by DNA protein kinase catalytic subunit association and subsequent DNA joining events[53]. Ku heterodimers are also constitutively associated with telomeres, where their tetramerization is suppressed by the TRF2 shelterin[53]. In this context, Ku70/Ku80 play an essential function in inhibiting the a-NHEJ and HR pathways, with loss of Ku resulting in chromosome fusions and loss of chromosome ends[11,14], which are similarly observed upon CSL knockdown. Despite its key function on telomeres, Ku tetramer association with chromosome ends may be suppressed by the t-loop structure, invoking other modes of telomere recruitment. Interestingly, Ku binding to telomeres in Saccharomyces cerevisiae is mediated by the Sir4 protein, which is tethered to telomeres via the Rap1 telomere binding protein[54]. Humans lack an obvious Sir4 homolog and our data suggest that CSL may provide an analogous function in mammals for the recruitment of Ku.

UPF1 is another pivotal determinant of genome stability, which ensures proper telomeric DNA replication through association of enzymes, such as DNA pol δ[19], and more structural components, such as the TPP1 shelterin[18]. UPF1 activity, rather than recruitment, appears to be controlled by TPP1[18]. While ATR-mediated phosphorylation was found to stabilize UPF1 localization at telomeres[19], its molecular anchoring partners were not yet established.

CSL is likely to play a general role in recruiting Ku70, Ku80, and UPF1 to telomeres, as the same CSL dependency was found in dermal fibroblasts and HEK293T cells, with both endogenous and over-expressed proteins. The dose-dependent increase of Ku70 binding to telomeres as a function of CSL indicates that

levels of the latter are rate limiting. In further interaction mapping studies, we found that the CSL BTD domain, which binds UPF1 and Ku70, is also the one anchoring them to telomeres. Specific CSL residues were found to be essential for Ku70 binding, which were instead dispensable for UPF1 association, suggesting that CSL may serve as a multifaceted anchoring scaffold for these and possibly other proteins to telomeres. It will be interesting to determine to what extent this function of CSL plays a role in other contexts of biological significance in which genomic stability is compromised, such as in various cancer types, aging tissues, and aging-related diseases[55].

## Methods

**Mice and human samples.** Characterization of mice with mesenchymal *Csl/Rbp-jκ* deletion and MDFs isolation was conducted as in[24]. All mouse work was performed according to the Swiss guidelines and regulations for the care and use of laboratory animals with approved protocol from the Canton de Vaud veterinary office (animal license No. 1854.4e) or was approved by the Massachusetts General Hospital Institutional Animal Care and Use Committee (MGH: 2004N000170). This study complies with all relevant ethical regulations.

Parts not needed for diagnosis of excised skin AK and SCC samples were obtained from the Departments of Dermatology, University of Zürich Biobank (Zurich) and Massachusetts General Hospital (Boston, MA), with institutional approvals and informed consent as part of institutional requirements (University Hospital Linz ECS 1119/2018 and Massachusetts General Hospital IRB No: 2018P003156). These included also tumor-free excess skin from routine Mohs surgery of skin cancer ("dog ears" resulting from bunching of skin at the end of an incision after wound closure). Experiments with these tissues were performed in adherence to the relevant ethical guidelines.

HDFs were prepared from discarded foreskin or abdominoplasty skin samples at the Department of Dermatology, Massachusetts General Hospital (Boston, Massachusetts, USA) with institutional approval (2000P002418), or were previously obtained[22]. Pairs of CAFs and matched HDFs from discarded skin SCC and flanking unaffected areas from the same (anonymized) patients, derived as in[28], were given specific identifiers as indicated in the different panels. CAF and matched HDF strains were used at very early passage of culturing (2nd–3rd passage). A list of cell strains is provided in Supplementary Data 3.

**Cell manipulations.** Conditions for culturing cells, viral shRNA infection, siRNA-mediated gene silencing, qPCR, RT-qPCR, and ChIP are as in[22,24,28]. HDF strains stably infected with a doxycycline-inducible lentiviral vector for MYC-tagged CSL in parallel with empty vector control[22] were treated for 5 days with doxycycline (500 ng ml$^{-1}$). Upon selection, HDFs were treated with a Bio-Link crosslinker UV irradiation system (Vilber Lourmat) equipped with a UVA lamp (375 nm), as indicated in the figure legends. A portable photometer IL1400A (International Light Technologies) was used for dosage determination. Samples were collected 6 h after UVA exposure. For the counteracting effects of CSL over-expression in HDFs with *CSL* gene silencing, HDF strains stably infected as above were treated with doxycycline (500 ng ml$^{-1}$) concomitantly to *CSL* silencing, as indicated in the figure legends. For the counteracting effects of CSL over-expression in CAFs, CAF strains were stably infected with a constitutive CSL over-expressing retrovirus or, upon infection with a doxycycline-inducible lentiviral vector for MYC-tagged CSL, were treated with increasing concentrations of doxycycline (50, 200, and 500 ng ml$^{-1}$) as indicated in the figure legends.

HDFs or CAFs were infected with lentiviruses and retroviruses as in[22]. All experiments were carried out with antibiotic resistance selection except for Supplementary Fig. 8b, c.

The siRNA and shRNA sequences used are provided in Supplementary Data 5 and 6. The oligonucleotides used in qPCR and ChIP are provided in Supplementary Data 4. The oligonucleotides used in RT-qPCR are provided in Supplementary Data 7. A detailed list of all the antibodies and the conditions used is in Supplementary Data 8.

**Immune detection and cell assays.** Immunofluorescence and immunoblots analyses were performed as in[22,24,28]. For immunofluorescence, cells were seeded on coverslips, fixed in 4% paraformaldehyde (PFA), and processed as in[28]. Immunohistochemistry of tumor and tissue sections was performed as in[22,24,56] and quantification of γ-H2AX, CSL, Ku70, Ku80, UPF1, TRF1, and TRF2 protein levels was made using ImageJ (NIH). Quantification of all other tissue immunofluorescence stainings was performed using ImageJ. Images were obtained with a Zeiss Observer Z1 inverted microscope and a Zeiss LSM880 confocal microscope. Antibodies used were anti-rabbit γ-H2AX antibody (Cat. 2577, Cell Signaling, 1:100 dilution), anti-goat VIMENTIN polyclonal antibody (Cat. AF2105, R&D, 1:200 dilution), anti-mouse VIMENTIN monoclonal antibody (Cat. 20346, Abcam, 1:200 dilution), anti-mouse CD45 monoclonal antibody (Cat. 304001, Biolegend, 1:200 dilution), anti-mouse CSL monoclonal antibody (Cat. 271128, Santa Cruz, 1:50 dilution), anti-rabbit UPF1 polyclonal antibody (Cat. HPA019587,

Sigma, 1:100 dilution), and anti-mouse PDGFRα-FITC monoclonal antibody (Cat. 21789, Santa Cruz, 1:50 dilution).

Unprocessed original scans of immunoblots are shown in Supplementary Figs. 10 and 11 and in the Source Data File. Antibodies used were anti-rabbit γ-H2AX antibody (Cat. 2577, Cell Signaling, 1:1000 dilution), anti-mouse γ-TUBULIN monoclonal antibody (Cat. GTU-88, Sigma, 1:2000 dilution), anti-rabbit CSL monoclonal antibody (Cat. 5313, Cell Signaling, 1:1000 dilution), anti-rabbit UPF1 monoclonal antibody (Cat. 109363, Abcam, 1:1000 dilution), anti-rabbit KU70 polyclonal antibody (Cat. 101820, GeneTex, 1:1000 dilution), anti-rabbit KU80 polyclonal antibody (Cat. 109935, GeneTex, 1:1000 dilution), and anti-rabbit Myc-tag monoclonal antibody (Cat. 2278, Cell Signaling, 1:2000 dilution).

Mean telomere length was measured using a quantitative PCR-based approach[30]. Briefly, total DNA was isolated from HDFs plus/minus *CSL* silencing using a DNeasy blood and tissue kit (Qiagen). DNA samples (20 ng) were loaded in triplicate in 20 μl reactions and run on a Light Cycler 480 II (Roche). Analysis of the qPCR output was performed using comparative quantification relative to the control sample.

**Alkaline comet assay.** Twenty microliters of cell suspension containing 10,000 cells were mixed with 130 μl of 0.5% low melting point agar, transferred onto microscope slides precoated with 1% normal melting point agarose, and incubated on ice for 10–15 min followed by coating with 0.5% low melting point agar. Cells were lysed in 2.5 M NaCl, 100 mM Na$_2$EDTA, 10 mM Tris-HCl, 1% Triton X-100, and pH > 13 for 2 h at 4 °C followed by 30 min incubation in 0.3 M NaOH and 1 mM EDTA, pH > 13 at 4 °C. Cells were electrophoresed for 25 min at 20 V, and were then treated three times with neutralizing buffer (0.4 M Tris, pH 7.5) for 5 min at 4 °C and stained with DAPI (Sigma-Aldrich). Images were obtained with Zeiss AxioImager Z1. Tail DNA % was calculated using Comet Score 1.6.1.13 software (www.rexhoover.com).

**Metaphase spread preparation.** Cells were collected for metaphase spread preparation after 2 h of 20 ng ml$^{-1}$ Colcemid (KaryoMAX™ Colcemid™ Solution, Gibco) treatment, trypsinized and swelled with pre-warmed 0.075 M KCl at 37 °C for 10 min. After centrifugation cell pellets were fixed with freshly made cold Carnoy's Fixative (3:1, methanol: glacial acetic acid). Cells were dropped onto glass slides, dried overnight, and stained with DAPI. For chromosome analysis metaphase spreads from random microscopic fields were counted.

**Telomere fluorescent in situ hybridization.** Telomere FISH was performed as in[57]. Briefly, metaphase spreads were rehydrated in PBS for 5 min at RT, fixed with 4% formaldehyde for 5 min, and dehydrated through an ethanol series (70, 90, and 100% ethanol for 5 min each). Chromosome spreads were hybridized to a telomeric PNA probe (F1013, Alexa Fluor 647, PNA Bio Inc.) resuspended in hybridization mix (10 mM Tris-HCl pH 7.4, 70% formamide, 0.5% blocking reagent) at 80 °C for 10 min followed by 3 h at room temperature. DNA was counterstained with DAPI. Images were obtained with Zeiss AxioImager Z1. For quantitative measurement of telomere length (Q-FISH), telomere fluorescence intensity was quantified using the TFL-TELO V2 software as in[58]. Telomere loss (OTL-TD) and chromosome fusions (SCF-EJ) were quantified manually.

**Meta-TIF analysis.** For combined immunofluorescence and telomere FISH, metaphase cells were treated for 2 h with 20 ng ml$^{-1}$ Colcemid, trypsinized and cell pellets were resuspended in 0.2% trisodium citrate, 0.2% KCl for 10 min at room temperature. Two hundred and fifty microliters of diluted cells were added to a cytofunnel (Thermo Scientific™ Shandon™ Double-Cytofunnel™) held in a cytoclip that had been pre-loaded with a Shandon Cytospin. Samples were spun for 10 min at 2000 r.p.m. with medium acceleration in a Shandon Cytospin® centrifuge. Cells were fixed with 3.7% formaldehyde for 10 min, permeabilized by potassium chromosome medium (KCM, 120 mM KCl, and 20 mM NaCl) for 10 min, and incubated with blocking solution (20 mM pH 7.5 Tris-HCl, 2% bovine serum albumin (BSA), and 150 mM NaCl, 0.1% Triton X-100) for 1 h. Samples were incubated with γ-H2AX antibody overnight at 4 °C followed by 1 h incubation with Alexa Fluor 488–conjugated secondary antibody. Cells were further processed following the telomere FISH protocol described above.

**TIF analysis in interphase nuclei.** Cells were seeded on coverslips and allowed to attach overnight. The following morning cells were fixed with 4% PFA for 10 min, permeabilized in 0.1% Triton X-100 for 10 min, and incubated with 5% BSA for 1 h. Samples were incubated with γ-H2AX antibody overnight at 4 °C followed by 1 h incubation with Alexa Fluor 594–conjugated secondary antibody. Cells were further processed following the telomere FISH protocol described above and hybridized to a telomeric PNA probe (F1009, FITC, and PNA Bio Inc.).

**Proximity ligation and immunoprecipitation assays.** Proximity ligation assays (PLAs)[59] were performed using Duolink PLA kit (Sigma) according to the manufacturer's protocol as in[28]. For PLA assays on clinical samples, tissues were fixed in 4% PFA, permeabilized in 0.1% Triton, and processed as in[28]. Images were

obtained with a Nikon Eclipse Ti confocal microscope. Antibodies used were anti-mouse CSL monoclonal antibody (Cat. 271128, Santa Cruz, 1:50 dilution), anti-rabbit UPF1 monoclonal antibody (Cat. 109363, Abcam, 1:50 dilution), anti-rabbit KU70 polyclonal antibody (Cat. 101820, GeneTex, 1:50 dilution), anti-rabbit KU80 polyclonal antibody (Cat. 109935, GeneTex, 1:50 dilution), anti-rabbit TRF1 polyclonal antibody (Cat. 32935, GeneTex, 1:50 dilution), and anti-mouse TRF2 monoclonal antibody (Cat. 13579, Abcam, 1:50 dilution).

Human Ku70 FL, 1-257, 1-464, 1-573, 464-609, and 257-609 cDNAs were generously provided by Dr Wen Yong Chen (The Rockefeller University, New York, USA), while human CSL FL, BTD, CTD, and NTD cDNAs were a kind gift of Dr. Franz Oswald (University Medical Center Ulm, Ulm, Germany). Mammalian expression vectors, expressing different CSL mutants (R192H, F235R, V237A, A258R, and Q307R), were generated by introducing either one or two point mutations in the human *CSL* coding sequence in order to mutate a single amino acid. The mammalian expression vector used for mutagenesis was pcDNA3. Flag-CSL expressing the full length sequence of the gene. Mutagenesis was perfomed by using the QuickChange Site-Directed Mutagenesis Kit (Agilent Technologies) following the manufacturer's instructions. The list of primers used for mutagenesis is provided in Supplementary Data 4.

Co-IPs were performed as in[22]. Co-IP assays in HEK293T cells transfected with CSL FLAG-tagged domains and Ku70 MYC-epitope tagged domains were carried out as in[22] using anti-FLAG M2 (Sigma) or anti-cMYC (Thermo Scientific) magnetic beads.

Re-immunoprecipitation experiments were performed as in[60]. Briefly, endogenous CSL was immunoprecipitated and the eluate was immunoprecipitated with anti-UPF1 antibody or non-immune IgG. Immunoprecipitates were analyzed by gel electrophoresis and immunoblotting with CSL, UPF1, Ku70, and Ku80 antibodies. Antibodies used were anti-rabbit CSL monoclonal antibody (Cat. 5313, Cell Signaling), and anti-rabbit UPF1 monoclonal antibody (Cat. 109363, Abcam).

**Microscale thermophoresis**. MST was performed as in[22]. Briefly, purified recombinant CSL (LabForce AG, Switzerland, TP602744), UPF1 (OriGene, USA, TP308018) and Ku70 (Origene, USA, TP304048) proteins were labelled with a RED-NHS protein labelling kit (NanoTemper). Labelled CSL protein was incubated at a constant concentration (1 μM) with two-fold serial dilutions of unlabeled UPF1 or Ku70 (from 9 μM to 0.27 nM) in standard MST buffer. Labeled UPF1 protein was incubated at a constant concentration (1 μM) with two-fold serial dilutions of unlabeled Ku70 (from 4 μM to 0.12 nM) in standard MST buffer. Equal volumes of proteins were mixed by pipetting and incubated at room temperature for 20 min. Telomere repeat duplex DNA $(TTAGGG)_3$ from Integrated DNA Technologies, labeled with 647-NHS, was incubated at a constant concentration (50 nM) with two-fold serial dilutions of unlabeled CSL (from 2 μM to 0.12 nM), TRF2 (from 700 to 0.02 nM), UPF1 (from 2 μM to 0.12 nM), and Ku70 (from 2 μM to 0.98 nM) in standard MST buffer. CSL specific motif duplex DNA GTTACTGTGGGAAAGAAAG and scrambled GCTACTCATACCTAGAACG (Microsynth AG) were incubated at two-fold serial dilutions (from 750 to 0.09 nM) with a constant concentration of labeled CSL (200 nM) in standard MST buffer. Mixtures were loaded in premium-treated glass capillaries (Monolith NT.115, MO-K025, NanoTemper) and loaded into the instrument (Monolith NT.115, Nano-Temper). Measurement protocol times were as follows: fluorescence before 5 s, MST on 30 s, fluorescence after 5 s, and delay 25 s. Analysis was performed at 40% light-emitting diode power and 60% laser power. The $K_d$ values were determined with the NanoTemper analysis tool.

**ChIP, Re-ChIP and ChIP-seq analysis**. For ChIP followed by dot blot analysis, immunoprecipitated DNA was processed and hybridized with radioactive probes as in[61]. Antibodies used were anti-rabbit CSL monoclonal antibody (Cat. 5313, Cell Signaling), anti-rabbit UPF1 monoclonal antibody (Cat. 109363, Abcam) anti-rabbit KU70 polyclonal antibody (Cat. 101820, GeneTex), anti-rabbit KU80 polyclonal antibody (Cat. 109935, GeneTex), anti-mouse Flag-M2 monoclonal antibody (Cat. F1804, Sigma), and anti-rabbit Myc-tag monoclonal antibody (Cat. 2278, Cell Signaling).

ChIP assays followed by quantitative PCR analysis were carried out as in[27] using CSL, UPF1, Ku70, Ku80, TRF1, and TRF2 antibodies. ChIP assays in HEK293T cells transfected with CSL FLAG-tagged vectors or Ku70 MYC-tagged construct were carried out as in[27] using FLAG-tag or MYC-tag antibodies, respectively.

Re-ChIP assays were performed by immunoprecipitation of endogenous CSL followed by immunoprecipitation of the eluate with UPF1 antibody as in[29]. Immunoprecipitates were analyzed by quantitative PCR analysis as in[27].

Raw data files from ChIP-seq assays (GSE59942)[22] were aligned to the GRCh38 genome with Bowtie2 Version 2.3.0 (http://bowtie-bio.sourceforge.net/bowtie2/index.shtml). Duplicates were removed with Picard (https://broadinstitute.github.io/picard/) and, for peak detection, MACS2 software (http://liulab.dfci.harvard.edu/MACS) was used with a p-value cutoff of 1.00e−04. Peaks were annotated with HOMER's (http://homer.ucsd.edu/homer/index.html) annotatePeaks.pl. Telomeric motif was created using HOMER's seq2profile.pl and CSL peaks were re-annotated using HOMER's annotatePeaks.pl. The Integrative Genomics Viewer (http://software.broadinstitute.org/software/igv/) was used for graphic illustration of ChIP-seq peaks.

**Telomerase activity assays**. Telomerase activity was measured using Telo-TAGGG™ Telomerase PCR ELISA kit (Roche) in accordance with the manufacturer's protocols. Briefly, cell protein extracts were prepared by lysing cultured CAFs or HDFs with cold lysis reagent provided with the kit. For each sample, 200 μg of total cell lysates were added to 25 μl reaction mixture containing the telomerase substrate, to a final the volume of 50 μl. Tubes were transferred to a thermal cycler and measurement protocol times were as follows: 25 °C for 30 min, 94 °C for 5 min, 30 cycles at 94 °C for 30 s, 50 °C for 30 s, and 72 °C for 90 s. Cell extracts that were previously heat-inactivated (at 95 °C for 10 min) were used as negative control, while HEK293T protein extracts supplied with the kit served as positive control. Upon dilution of 5 μl PCR product with the Hybridization buffer, 100 μl of mixture were added onto each well precoated with a digoxigenin-labeled telomeric repeat–specific probe. Following denaturation and hybridization at 37 °C for 2 h, anti-digoxigenin peroxidase conjugate and TMB substrate were used for the ELISA assay, and the absorbance at 450 nm (with a reference wave length of 690 nm) was measured using an LEDETECT 96 microplate reader (Dynamica, Germany). Based on the values of the negative controls, samples with absorbance values less than 0.25 were considered negative.

**LCM experiments**. Skin and SCC frozen samples used for LCM followed by RT-qPCR were provided by the Department of Dermatology, Massachusetts General Hospital (Boston, Massachusetts, USA), with institutional review boards approvals and informed consent. LCM was performed using an Arcturus XT microdissection system (Applied Biosystems) as in[22,24]. The oligonucleotides used in RT-qPCR are provided in Supplementary Data 7.

**Docking Analysis**. Atomic coordinates of CSL, telomere DNA duplex, and Ku70/Ku80 were extracted from the crystallography structures available in the RCSB protein data bank with access codes PDB ID: 3V79 [10.2210/pdb3V79/pdb], PDB ID: 4J19 [10.2210/pdb4J19/pdb], and PDB ID: 1JEQ [10.2210/pdb1JEQ/pdb], respectively. The missing residues of Ku70 loops were modeled by using Swiss model server (https://swissmodel.expasy.org/). First, to engender the 3D model of CSL-telomeric DNA-Ku70/Ku80, docking between CSL and telomere DNA was performed by using HDOCK server (http://hdock.phys.hust.edu.cn/), which uses a FTT-based docking algorithm. The top ranked 3D models for CSL-telomeric DNA showing less deviation as compared with CSL-DNA motif were further selected. Second, CSL-telomeric DNA models were docked with Ku70/Ku80 by using HDOCK server. On the basis of the mutagenesis data, CSL-telomeric DNA-Ku70/Ku80 3D models were ranked and the top hit model was further analyzed. PyMOL 2.3 software by Schrodinger (https://pymol.org/2/) was used to generate all docking images and to create the movie.

**Transcriptome analysis**. Two strains of HDFs were infected with two different shRNAs against *UPF1* in parallel with a control virus. Total RNA was extracted 7 days post-infection (including 5 days of antibiotic resistance selection to elim-inate uninfected cells) using the directZol RNA miniprep kit (Zymo Research) with on-column DNase treatment. RNA quality was verified on the Bioanalyzer (Agilent Technologies) with RNA integrity number (RIN) > 8. A total of 500 ng of total RNA were depleted of ribosomal RNA using Ribo-zero RNA removal kit (Illumina) and were used for library preparation using a Truseq kit (Illumina). A single read analysis was performed on the IlluminaHiSeq 2000 sequencer at the Genomic Technologies Facility (Lausanne University). Reads were trimmed using Trim-momatic (v0.22) and then mapped to the human hg19 reference genome using TopHat (v2.0.8b). Gene expression levels were evaluated using the HTSeq package (release 0.5.4p1).

Duplicate cultures of an HDF strain were infected with two different shRNAs against *CSL* in parallel with a control virus. Total RNA was extracted 7 days post-infection as above. A total of 1 μg of total RNA was used for library preparation using a NEBNext Ultra DNA Library Preparation kit (New England Biolabs). Gene expression analysis was performed as above.

RNA-seq data are deposited in the public repository (GSE113542). List of genes differentially expressed upon *CSL* or *UPF1* silencing are provided in Supplementary Data 1.

The list of genes significantly regulated by *CSL* or *UPF1* silencing are in Supplementary Data 1.

**Statistical analysis**. Data are presented as mean ± SEM or mean ± SD or ratios among treated and controls, with two to three separate HDF or CAF strains in independent experiments as indicated in the figure legends. For genomic analysis and functional testing assays, statistical significance of differences between experimental groups and controls was assessed by one-way ANOVA, two-tailed unpaired or paired t-test. P values < 0.05 were considered as statistically significant. The researchers were not blinded and no strain or result was excluded from the analysis.

**Reporting summary**. Further information on research design is available in the Nature Research Reporting Summary linked to this article.

## Data availability

The RNA-seq and ChIP-seq data for this study are deposited in GEO with the accession codes GSE113542 and GSE59942, respectively. All other relevant data generated in this paper that support the findings of this study are available upon request from the authors. The source data underlying Figs. 1a–g, 2a–f, 3a–g, 4a–d, 5a–e, 6c, d, 7a–f, 8a–d, and g–j and Supplementary Figs. 1b–d, 3a–d, 4a, c, d, 6a–f, 7b, 8a–f, and 9a are provided as a Source Data file.

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

## Acknowledgements

We are grateful to Drs C. Laszlo and S. Jo for preliminary MS and Co-IP analysis, respectively. We thank Dr P. Ostano for bioinformatic support. We are grateful to Dr. Wen Yong Chen and Dr Franz Oswald for Ku70 and CSL cDNA constructs, respectively. This study was supported by grants from the Swiss National Science Foundation (310030B_176404 "Genomic instability and evolution in cancer stromal cells"), the European Research Council (26075083), and the NIH (R01AR039190, R01AR064786; the content does not necessarily represent the official views of the NIH) to GPD.

## Author contributions

G.B., A.K., B.T., So.G., A.C., Sa.G., P.B. and P.J. performed the experiments and/or contributed to analysis of the results. F.T. performed the ChIP-seq bioinformatics analysis. T.L. and J.L. contributed to telomere ChIP dot blot experiments. W.H. and V.N. provided clinical samples. G.B. and G.P.D. designed the study and wrote the paper.

## Additional information

**Competing interests:** The authors declare no competing interests.

