## [Peer Review File · Nature Communications]

Reviewers' comments:

Reviewer #1 (Remarks to the Author):

I thank the authors for their thoughtful response and for experimentally addressing my concerns. I agree with the decision to focus on the CSL/telomere part. Considering the response to all three referees I am supportive of publication in the current form.

Jan Karlseder

Reviewer #2 (Remarks to the Author):

In this second round of revision for the manuscript from Bottoni et al., the overall quality of the paper has been much improved. The second version of the manuscript is now focusing extensively on the role of CSL loss in the genomic instability in the carcinoma-associated fibroblasts from skin neoplasia. This manuscript describes that loss of CSL expression, linked to carcinoma-associated fibroblast initiation, promotes genomic instability (detection of histone gamma-H2AX phosphorylation positive staining and telomeres length, loss of one or two telomeres). In addition, in this second version of the manuscript the authors emphasized strongly the interaction of CSL with telomeric sheltering complex proteins with a strong and convincing biochemistry analysis.

It is very appreciable that the authors answered most of the reviewers concerns in their rebuttal letter.

However, I still notice some caveats within the second version of the manuscript that need to be taken in consideration by the authors.

As stated in the first round of revision, overall, the quality of the imaging remain low and it is therefore difficult to appreciate some immunostaining co-localisation, specifically in F1a and F2a/2b. Looking at this images, I notice that first, the rectangles in the upper images (low magnification) do not correspond to the lower image (higher magnification), second, I am still not convinced that the gamma-H2ax positive cells are also the vimentin positive. To my point of view, the images are not in accordance with the quantification provided in F1a and in F2a/2b. Along with this, the choice of the yellow color to stain the keratin positive cells in F2a is misleading. Indeed, the co-localisation of the gamma-H2ax (red) and vimentin (green) positive cells also results in yellow colour.

This manuscript highlights a strong genomic instability in carcinoma-associated fibroblasts, either obtained by silencing of CSL expression in fibroblasts or in carcinoma-associated fibroblasts isolated from tumor biopsies. Accumulation of genetic alterations in a such extend should lead to either cell death, cell cycle arrest or cell transformation. However, CAF, either obtained in vitro or through biopsies isolation, are growing in vitro and are not transformed (they do not induce tumor when injected in mice). How the authors explain this aspect? Could the authors provide explanation and discuss this aspect in the discussion section?

Moreover, I have notice that most of the mice experiment presented in F1 and F2 rely on a very low number of specimens (n(WT P0)=1; WT p9=2; KO P0=2 and WT P9 (probably an error here)=2), only 5 specimens of AK were analysed. Those numbers are to low for statistic.

CSL loss have been shown in previous publication, from the same research group, to play a significant role in CAF initiation. Moreover, multiple research groups (it is not a debate in the literature) provided strong evidence that TGFbeta (and other growth factors and cytokines) or cancer cell conditioned media initiate CAF in tumor. What is the effect of TGFbeta and tumor conditioned media on genomic instability in fibroblast activation? If TGFbeta and cancer conditioned media induce genomic instability in fibroblast, would it be mediated by CSL?

Answers to Reviewers' comments:

Reviewer #1:

I thank the authors for their thoughtful response and for experimentally addressing my concerns. I agree with the decision to focus on the CSL/telomere part. Considering the response to all three referees I am supportive of publication in the current form.

Jan Karlseder

Authors' Response: We are very thankful to the Reviewer for the kind and favorable comments. We very much appreciate his expert opinion and constructive suggestions, which helped us to improve the quality of the manuscript.

Reviewer #2:

In this second round of revision for the manuscript from Bottoni et al., the overall quality of the paper has been much improved. The second version of the manuscript is now focusing extensively on the role of CSL loss in the genomic instability in the carcinoma-associated fibroblasts from skin neoplasia. This manuscript describes that loss of CSL expression, linked to carcinoma-associated fibroblast initiation, promotes genomic instability (detection of histone gamma-H2AX phosphorylation positive staining and telomeres length, loss of one or two telomeres). In addition, in this second version of the manuscript the authors emphasized strongly the interaction of CSL with telomeric sheltering complex proteins with a strong and convincing biochemistry analysis. It is very appreciable that the authors answered most of the reviewers concerns in their rebuttal letter.

However, I still notice some caveats within the second version of the manuscript that need to be taken in consideration by the authors.

Authors' Response: We thank the Reviewer for finding the paper of interest and for the constructive suggestions. As recommended by the Reviewer, we have:

1) Improved the quality of the immunofluorescence images in Fig. 1a and Fig. 2a,b by replacing them with those obtained by confocal microscopy.

2) Increased the number of mouse tissues and Actinic Keratosis samples analyzed to a total number of 18 mice (9 WT + 9 Cs/ KO) and 11 AK lesions from 11 patients, respectively (in Fig. 1a and 2a), thereby improving the statistical significance of the findings.

3) Expanded the discussion, to consider possible mechanisms that restrain CAF transformation in spite of their genomic instability and to mention the connection between TGF β signaling and genomic instability.

Specific comments:

1. As stated in the first round of revision, overall, the quality of the imaging remain low and it is therefore difficult to appreciate some immunostaining co-localisation, specifically in F1a and F2a/2b. Looking at this images, I notice that first, the rectangles in the upper images (low magnification) do not correspond to the lower image (higher magnification), second, I am still not convinced that the gamma-H2ax positive cells are also the vimentin positive. To my point of view, the images are not in accordance with the quantification provided in F1a and in F2a/2b. Along with this, the choice of the yellow color to stain the keratin positive cells in F2a is misleading. Indeed, the co-localisation of the gamma-H2ax (red) and vimentin (green) positive cells also results in yellow colour.

We apologize for the lack of clarity in presenting our experimental results. We have improved the quality of the IF images shown in Fig. 1a and Fig. 2a,b, and we have indicated more clearly in the figure legends which areas of the different lesions were shown at higher magnification. Regarding the concerns on the yellow color used in our previous immunofluorescence, we have performed novel experiments adding 6 additional AK samples using only three channels (magenta, cyan and blue staining γ -H2AX, Vimentin and nuclei respectively).

2. This manuscript highlights a strong genomic instability in carcinoma-associated fibroblasts, either obtained by silencing of CSL expression in fibroblasts or in carcinoma-associated fibroblasts isolated from tumor biopsies. Accumulation of genetic alterations in a such extend should lead to either cell death, cell cycle arrest or cell transformation. However, CAF, either obtained in vitro or through biopsies isolation, are growing in vitro and are not transformed (they do not induce tumor when injected in mice). How the authors explain this aspect? Could the authors provide explanation and discuss this aspect in the discussion section?

We thank the Reviewer for the very interesting question, which we have addressed in the discussion section. As rightfully stated by the Reviewer, accumulation of genetic alterations leads to a plethora of consequences, including cell death or cell cycle arrest as failsafe mechanisms, or cell transformation. Although genetic instability has been defined as a hallmark of cancer, recent studies have shed light on the role of genomic abnormalities in healthy tissues¹, broadening our understanding of tumor development and aging. As we now point out in the discussion (page 17 line 5), we can speculate that the lack of a transformed phenotype in CAFs might depend on multiple aspects:

1) CAFs do not show a high proliferative potential in comparison to other cell types, such as epithelial cells, which are constantly subjected to renewal. The limited cell proliferation rate could therefore account for the containment of cells carrying genetic abnormalities.

2) Cell transformation is driven by the accumulation of multiple aberrations, which need to co-occur in order to establish a selective advantage.

3) As compared to cancer cells, multiple suppressive mechanisms still take place in CAFs, therefore limiting clonal growth.

4) CAFs, as well as normal cells, are largely diploid, with a limited number of copy-number changes, while cancers typically display extensive aneuploidy.

3. Moreover, I have notice that most of the mice experiment presented in F1 and F2 rely on a very low number of specimens ($n(\text{WT } P0)=1$; $\text{WT } p9=2$; $\text{KO } P0=2$ and $\text{WT } P9$ (probably an error here)=2), only 5 specimens of AK were analysed. Those numbers are to low for statistic.

We thank the Reviewer for the constructive suggestions on how to improve our data. To further demonstrate the statically significant increase in γ -H2AX levels in *Csl* KO versus WT mice, and in HDFs from clinical AK lesions versus fibroblasts of flanking unaffected skin, we have increased the number of samples and quantified IF signals (Fig. 1a and Fig. 2a).

CSL loss have been shown in previous publication, from the same research group, to play a significant role in CAF initiation. Moreover, multiple research groups (it is not a debate in the literature) provided strong evidence that TGFbeta (and other growth factors and cytokines) or cancer cell conditioned media initiate CAF in tumor. What is the effect of TGFbeta and tumor conditioned media on genomic instability in fibroblast activation? If TGFbeta and cancer conditioned media induce genomic instability in fibroblast, would it be mediated by CSL?

We thank the Reviewer for the very interesting question. As indicated by the reviewer and as we now point out in the discussion (page 17 line 13): "TGF β signaling, which functions as an important trigger of CAF activation^{2,3}, has been reported to modulate genomic instability in a variety of cell types, including normal fibroblasts, through unrepaired DNA strand breaks and differential expression of DDR genes^{4,5}. While TGF β treatment of dermal fibroblasts has little or no effects on CSL expression (our unpublished observations), silencing of the latter elicits global changes in gene expression that overlap, in part, with those triggered by TGF β ^{6,7}. Modulation of DDR genes and other indirect mechanisms related to the CSL transcription regulatory function might play a relevant role in preserving genomic stability. However, separate from transcription, we have shown that the CSL protein directly participates in maintenance of chromosomal ends, as part of a telomere-protective complex comprising the Ku70/Ku80 and UPF1 proteins".

We note a recent report showing TGF β induction upon telomere shortening, which enhances idiopathic pulmonary fibrosis. The findings suggest that TGF β might be a target of telomere-driven genomic instability instead of being causative⁸. Irrespectively, a possible interplay between CSL and signaling by TGF β - and/or other cytokines - in CAF activation and genomic instability is an interesting possibility for future studies.

- 1 Martincorena, I. *et al.* Tumor evolution. High burden and pervasive positive selection of somatic mutations in normal human skin. *Science* **348**, 880-886, doi:10.1126/science.aaa6806 (2015).
- 2 Gascard, P. & Tlsty, T. D. Carcinoma-associated fibroblasts: orchestrating the composition of malignancy. *Genes Dev* **30**, 1002-1019, doi:10.1101/gad.279737.116 (2016).
- 3 Kalluri, R. The biology and function of fibroblasts in cancer. *Nat Rev Cancer* **16**, 582-598, doi:10.1038/nrc.2016.73 (2016).
- 4 Legrand, A. J. *et al.* Persistent DNA strand breaks induce a CAF-like phenotype in normal fibroblasts. *Oncotarget* **9**, 13666-13681, doi:10.18632/oncotarget.24446 (2018).

- 5 Barcellos-Hoff, M. H. & Cucinotta, F. A. New tricks for an old fox: impact of TGFbeta on the DNA damage response and genomic stability. *Sci Signal* **7**, re5, doi:10.1126/scisignal.2005474 (2014).
- 6 Procopio, M. G. *et al.* Combined CSL and p53 downregulation promotes cancer-associated fibroblast activation. *Nat Cell Biol* **17**, 1193-1204, doi:10.1038/ncb3228 (2015).
- 7 Kim, D. E. *et al.* Convergent roles of ATF3 and CSL in chromatin control of cancer-associated fibroblast activation. *J Exp Med*, doi:10.1084/jem.20170724 (2017).
- 8 Liu, Y. Y., Shi, Y., Liu, Y., Pan, X. H. & Zhang, K. X. Telomere shortening activates TGF-beta/Smads signaling in lungs and enhances both lipopolysaccharide and bleomycin-induced pulmonary fibrosis. *Acta Pharmacol Sin* **39**, 1735-1745, doi:10.1038/s41401-018-0007-9 (2018).